# DIFFICULTY–DIVERSITY COLLABORATIVE FILTERING FOR DATA-EFFICIENT LLM FINE-TUNING

**Long P. Hoang[1]**    **Wenxuan Zhang[1]**    **Wei Lu[2]**
[1]Singapore University of Technology and Design    [2]Nanyang Technological University
`long_hoang@mymail.sutd.edu.sg`
`wxzhang@sutd.edu.sg, wei.lu@ntu.edu.sg`

## ABSTRACT

The performance of fine-tuned language models is heavily influenced by the quality and quantity of their fine-tuning data. While scaling laws suggest that larger models benefit from more data during pretraining, the Less-is-More hypothesis highlights that downstream fine-tuning often requires only a small but high-quality dataset to effectively elicit a model's pretrained knowledge. However, identifying such premium data, particularly in terms of difficulty and diversity, typically relies on human expertise, and existing methods offer limited guidance for automatic selection from large unannotated corpora. This work presents a novel quantitative framework that formalizes the interplay between question difficulty and diversity, and introduces *Difficulty–Diversity Collaborative Filtering* (DDCF): an automated approach that tailors data selection to the unique characteristics of each language model via collaborative filtering. By leveraging a small seed dataset to predict correctness across a large unannotated corpus, our method reduces the annotation cost by $100-200\times$, while maintaining downstream performance comparable to full-corpus fine-tuning.[1]

## 1 INTRODUCTION

The remarkable success of Large Language Models (LLMs) in recent years (Grattafiori et al., 2024b; Yang et al., 2025b) stems largely from their ability to learn rich and generalizable representations from massive pretraining corpora. To further enhance capabilities of these models on downstream tasks, supervised fine-tuning (SFT) has become a popular approach (Wei et al., 2022; Chung et al., 2024). However, SFT typically involves fine-tuning pretrained models on large-scale, human-annotated instruction datasets, often comprising hundreds of thousands of examples.

Despite its effectiveness, fine-tuning on such large datasets presents several challenges. First, data collection and model training incur substantial computational costs. Second, updating a model on a new large corpus may cause catastrophic forgetting, where continual learning of new tasks degrades performance on previously acquired knowledge (Biderman et al., 2024; Wang et al., 2024). Third, scaling up the dataset often leads to over-representation of common patterns, reducing diversity and underrepresenting rare but important examples (Kim et al., 2022; Zhang et al., 2025a).

Recently, the *Less-is-More* hypothesis (Zhou et al., 2023; Ye et al., 2025; Dohmatob et al., 2025) has suggested that downstream task adaptation can be achieved through minimal supervision, where the model primarily learns task-specific formatting or styles to reveal knowledge already encoded during pretraining. Empirical studies have shown that fine-tuning on just a few carefully selected examples sometimes outperforms naively using vast annotated corpora (Zhou et al., 2023; Ye et al., 2025; Muennighoff et al., 2025). Furthermore, theoretical analysis (Dohmatob et al., 2025) demonstrates that, when the base model is strong, selecting harder examples offers a provable advantage. However, such curated datasets often rely on evolving human expertise, making them labor-intensive, inflexible, and inconvenient to adapt to new models or tasks.

While recent efforts have explored automated methods to improve data quality (Xia et al., 2024; Yang et al., 2024b), the automatic selection without annotated output responses remains an open

---

[1]Our code is publicly available at `https://github.com/iNLP-Lab/DDCF`.

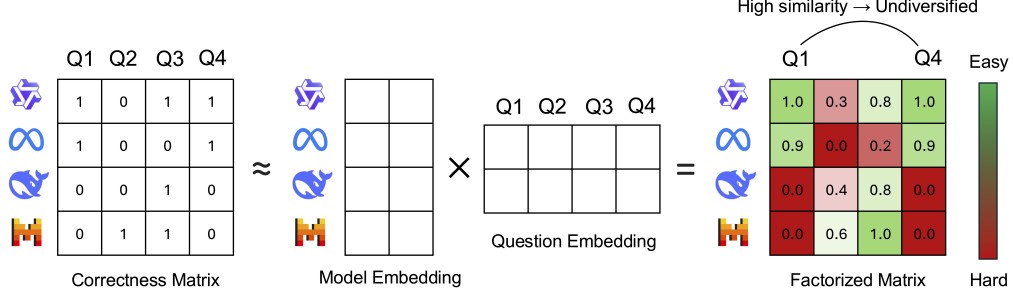

Figure 1: An illustration of our proposed *Difficulty–Diversity Collaborative Filtering* framework. Given a dataset, a binary correctness matrix from model predictions is factorized into model and question embeddings. Difficulty is defined by factorized scores, while diversity is measured by question similarity. These two criteria jointly guide the selection of compact yet effective training subsets, providing strong learning signals while avoiding redundant, overly similar examples.

challenge. For example, (Xia et al., 2024) leveraged gradient matching to a target dataset, while (Yang et al., 2024b) instead trained the LM on the entire annotated corpus and then selected samples by clustering their loss trajectories. Other approaches (Ye et al., 2025; Muennighoff et al., 2025; Marion et al., 2023) identify challenging examples based on binary correctness, reasoning length, or perplexity, and group them into manually defined categories. However, such approaches are not universal—questions deemed difficult for one model may not be difficult for another, and rarely achieve an optimal balance of difficulty and diversity.

To address these gaps, we propose *Difficulty–Diversity Collaborative Filtering* (DDCF), a framework that reduces both annotating and fine-tuning costs by automatically selecting a compact, high-quality subset of questions tailored to each target model. As illustrated in Figure 1, DDCF measures question difficulty using collaborative filtering over predicted correctness patterns from multiple open-source LLMs, and quantifies diversity based on question similarity. By formulating data selection as a combinatorial optimization that directly trades off these two criteria, we can efficiently approximate the optimal subset using a simple *k-greedy* strategy. Starting from an empty set, k-greedy iteratively adds the question with the greatest marginal gain in our difficulty–diversity objective until exactly $k$ examples are selected. Empirically, DDCF selects compact yet impactful subsets that effectively challenge the target model while maintaining broad coverage, enabling more efficient fine-tuning and improved performance in various downstream tasks.

Our key contributions are as follows:

- We propose *Difficulty–Diversity Collaborative Filtering*, a novel framework that leverages multiple LMs to capture unique characteristics of each target LM, enabling automatic construction of compact, model-specific training subsets *without requiring prior annotation*.

- To the best of our knowledge, this work is the first to systematically quantify and analyze the interplay between difficulty and diversity in data selection, and to demonstrate how their trade-off shapes downstream fine-tuning performance.

- We empirically demonstrate that DDCF outperforms existing data selection baselines across multiple benchmarks, achieving higher accuracy with the same selection budget.

## 2   RELATED WORK

Numerous approaches have been proposed to curate high-quality training data, which can be grouped into several categories. Influence-based methods estimate each example's impact on a target set via gradient matching—e.g., Grad-Match (Killamsetty et al., 2021), LESS (Xia et al., 2024), NICE (Wang et al., 2025)—or by framing selection as an optimal control problem (Gu et al., 2025). Heuristic approaches frequently utilize token probability statistics as a proxy for difficulty, exemplified by strategies prioritizing medium-perplexity (Marion et al., 2023) or analyzing training dynamics via

Table 1: Comparison of DDCF with prior data selection methods. "Difficulty-Aware" and "Diversity-Aware" reflect whether these criteria are considered in selection. "No Full-Corpus Fine-Tuning" indicates whether the method avoids training on the full corpus. "Minimal-Annotation" denotes methods that (almost) do not rely on annotations, thereby reducing annotation costs. "No LLM Feedback" indicates the method does not depend on external reward models, e.g,. ChatGPT.

| Method | Difficulty-Aware | Diversity-Aware | No Full-Corpus Fine-Tuning | Minimal Annotation | No LLM Feedback |
|---|---|---|---|---|---|
| Perplexity (Marion et al., 2023) | ✓ | ✗ | ✓ | ✗ | ✓ |
| S2L (Yang et al., 2024b) | ✓ | ✓ | ✗ | ✗ | ✓ |
| AlpaGasus (Chen et al., 2024) | ✓ | ✗ | ✓ | ✗ | ✗ |
| LESS (Xia et al., 2024) | ✓ | ✓ | ✓ | ✗ | ✓ |
| DiSF (Fan et al., 2025) | ✗ | ✓ | ✓ | ✓ | ✓ |
| **DDCF (ours)** | ✓ | ✓ | ✓ | ✓ | ✓ |

Dataset Cartography (Swayamdipta et al., 2020). Feedback-driven frameworks leverage closed-source LLMs (such as ChatGPT) to score or prune candidates—exemplified by AlpaGasus (Chen et al., 2024) and Evol (Liu et al., 2024). Diversity-aware sampling ensures broad representational coverage through embedding similarity (e.g., D4 (Tirumala et al., 2023), DiSF (Fan et al., 2025)), while lightweight proxy models cluster examples from loss trajectories, as in S2L (Yang et al., 2024b). Datamodels (Ilyas et al., 2022; Chang & Jia, 2023) estimate how the presence of each training/few-shot example affects the target model, but its combinatorial formulation ignores example semantics and thus cannot generalize beyond the observed training questions.

Parallel to data selection, recent work has explored the problem of choosing the most appropriate model for a given question, commonly referred to as *LLM routing*. FrugalGPT (Chen et al., 2023) adaptively queries models in sequence until a reliable answer is obtained. More recent methods embed models and questions into a shared latent space and learn routing policies using matrix factorization (Ong et al., 2024; Zhuang et al., 2025), while Nguyen et al. (2024) frame the problem as a multi-armed bandit.

Our work bridges these two lines of research by recasting model–question interactions as a recommendation problem (Lee & Seung, 2000; He et al., 2017), treating models as users and questions as items. This perspective allows us to learn tailored relevance scores that guide data selection in a large corpus, even without full-annotation labels. Building on this, we propose a lightweight collaborative filtering framework with difficulty–diversity re-ranking to curate a small, high-quality subset from a large unannotated corpus, yielding strong performance in low-resource fine-tuning.

Table 1 summarizes how DDCF compares to representative data selection approaches across five key dimensions. Notably, DDCF only relies on ground-truth answers from a small seeding dataset to construct the binary correctness matrix. This design enables DDCF to uniquely satisfy all five criteria: it selects a compact, challenging, and diverse subset without the need for full-corpus fine-tuning, external annotations, or feedback from closed-source LLMs. As a result, DDCF offers a scalable and domain-agnostic solution for efficient data curation across diverse model families. Given the convenience of DDCF and its potential for widespread use, the framework can readily extend beyond supervised fine-tuning to settings such as In-Context Learning (Chang & Jia, 2023; Li & Qiu, 2023; Pecher et al., 2024; Purohit et al., 2024; 2025), Active Learning (Zhang & Plank, 2021), and even Curriculum Learning (Soviany et al., 2022).

## 3  DATA SELECTION WITH MINIMAL ANNOTATION

### 3.1  CORRECTNESS PREDICTOR

Given $m$ language models $\mathcal{M} = \{\mathcal{M}_1, \ldots, \mathcal{M}_m\}$ and a seed dataset of $n$ questions $\mathcal{Q} = \{q_1, \ldots, q_n\}$ with corresponding ground-truth answers, we construct a binary correctness matrix $\mathcal{A} \in \{0, 1\}^{m \times n}$. Each entry $\mathcal{A}_{ij}$ indicates whether model $\mathcal{M}_i$ correctly answers question $q_j$. This matrix captures fine-grained model-question interactions, enabling us to analyze both model capa-

bilities and question difficulty. For instance, certain models may perform well on algebra but poorly on geometry, while questions answered incorrectly by most models likely indicate higher difficulty.

Following the approach in Zhuang et al. (2025), we enrich the binary matrix $\mathcal{A}$ by learning low-dimensional embeddings for both models and questions. Specifically, we learn model embeddings $E_M \in \mathbb{R}^{m \times d}$ and question embeddings $E_Q \in \mathbb{R}^{n \times d}$ such that

$$\mathcal{A} \approx \hat{\mathcal{A}} = E_M E_Q^\top, \tag{1}$$

where $d$ denotes the embedding dimension and $\hat{\mathcal{A}}$ approximates the observed correctness matrix $\mathcal{A}$. This factorization is analogous to those used in collaborative filtering (Lee & Seung, 2000; He et al., 2017), but it is inherently limited to the training set and does not generalize to unseen questions.

To enable generalization, we introduce a correctness predictor $f : \mathcal{M} \times \mathcal{Q} \to [0, 1]$, which estimates whether a given model correctly answers a given question. We instantiate $f$ using an encoder architecture, detailed below.

**Encoder**   The encoder comprises two modules: a model encoder $\phi_M$ and a question encoder $\phi_Q$, both projecting into a shared latent space $\mathbb{R}^d$.

The model encoder $\phi_M : \mathcal{M} \to \mathbb{R}^d$ maps a model index to an initial representation.

The question encoder $\phi_Q : \mathcal{Q} \to \mathbb{R}^d$ is defined as a function composition $\phi_Q = h_Q \circ g_Q$, where:

- $g_Q : \mathcal{Q} \to \mathbb{R}^{\dim_q}$ uses a pre-trained sentence transformer to encode question text into an initial question representation $E_{q_j}^0$;

- $h_Q : \mathbb{R}^{\dim_q} \to \mathbb{R}^d$ projects this representation into the shared latent space, yielding factorized question embeddings $E_{q_j}$ for each $q_j \in \mathcal{Q}$.

In our implementation, $h_Q$ is a multilayer perceptron.

**Classifier Head**   The classifier predicts correctness from the Hadamard product of embeddings:

$$\psi(E_{m_i} \odot E_{q_j}),$$

where $\psi : \mathbb{R}^d \to \mathbb{R}^2$ is a linear classifier. The overall predictor is thus defined as $f(\mathcal{M}_i, q_j) = \psi(\phi_M(\mathcal{M}_i) \odot \phi_Q(q_j))$, which can be trained using binary cross-entropy loss.

The predicted correctness score for model $\mathcal{M}_i$ on question $q_j$ is defined as:

$$\hat{\mathcal{A}}_{ij} = \sigma(f(\mathcal{M}_i, q_j)_1), \tag{2}$$

where $\sigma(\cdot)$ is the sigmoid function, and the subscript 1 denotes the logit for the "correct" class.

Notably, Equation 2 can be viewed as a parameterized version of the classical matrix factorization in Equation 1. Instead of estimating a single shared difficulty score per question, this formulation allows the difficulty of a question to be *"personalized"* for each model's characteristic. This personalized modeling of correctness underpins our approach in the next section, where we construct *Difficulty–Diversity Collaborative Filtering* strategies tailored to individual models.

## 3.2   DIFFICULTY-DIVERSITY COLLABORATIVE FILTERING

Given a target model $\mathcal{M}_i$ and a *large unannotated dataset* $\mathcal{D}$, where $|\mathcal{D}| \gg |\mathcal{Q}|$ and $\mathcal{Q}$ is the introduced seed dataset, our goal is to select a subset $S_i \subset \mathcal{D}$ of $k$ questions that are both (1) *difficult* for the model $\mathcal{M}_i$ and (2) *diverse* to cover a wide range of topics. This ensures that the selected examples provide strong learning signals while avoiding redundancy.

To estimate question difficulty, we leverage the correctness predictor $f$ introduced earlier. For every question $q_j \in \mathcal{D}$, the model $\mathcal{M}_i$'s predicted correctness score is given by $\tilde{\mathcal{A}}_{ij} = \sigma(f(\mathcal{M}_i, q_j)_1)$, and we aggregate these into a vector $\tilde{\mathcal{A}}_i \in \mathbb{R}^{|\mathcal{D}|}$. Lower values of $\tilde{\mathcal{A}}_{ij}$ correspond to questions the model is more likely to get wrong, thus indicating higher difficulty.

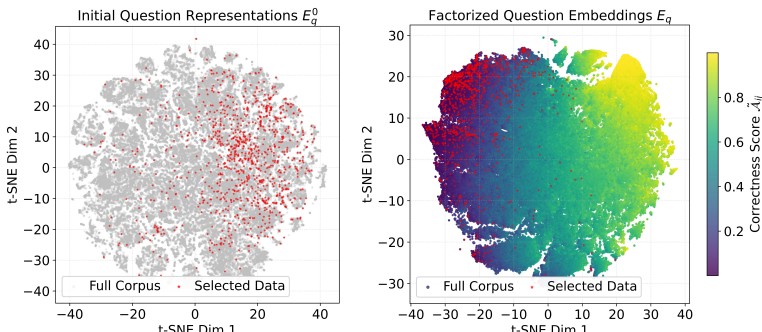

Figure 2: t-SNE visualization of questions selected by DDCF (best viewed in color). **Left:** the selected data in the semantic space encoded by Sentence-Transformer. **Right:** the same data in the factorized space learned by the correctness predictor, with point color indicating difficulty (darker means harder) and selected examples highlighted in red. Our proposed DDCF framework organizes the subset to target challenging questions while preserving diversity across semantic regions.

To encourage diversity among selected questions, we define a similarity matrix $\Sigma \in \mathbb{R}^{|\mathcal{D}| \times |\mathcal{D}|}$, where each entry is the cosine similarity of the sentence-transformer embeddings of questions $q_u$ and $q_j$:

$$\Sigma_{uj} = \frac{(E_{q_u}^0)(E_{q_j}^0)^\top}{\|E_{q_u}^0\|_2 \|E_{q_j}^0\|_2}. \tag{3}$$

We then formulate the selection problem as a combinatorial optimization task:

$$\min_{x \in \{0,1\}^{|\mathcal{D}|}} \lambda(x^\top \tilde{\mathcal{A}}_i) + (1-\lambda)(x^\top \Sigma x), \quad \text{s.t.} \sum_{j=1}^{|\mathcal{D}|} x_j = k. \tag{4}$$

Here, the constraint enforces the selection of exactly $k$ questions, forming $S_i = \{q_j | x_j = 1\}$, i.e., the curated subset for the model $\mathcal{M}_i$, and $\lambda \in [0,1]$ balances the trade-off between difficulty $(x^\top \tilde{\mathcal{A}}_i)$ and diversity $(x^\top \Sigma x)$. Although the objective is convex over the continuous relaxation of $x$, the binary constraint renders the problem NP-hard and computationally intensive due to the $O(|\mathcal{D}|^2)$ memory complexity of the similarity term $\Sigma$.

To overcome these limitations, we propose a memory- and compute-efficient $k$-greedy heuristic that incrementally selects questions. Starting with an empty set $S_i$, at each step, we add the next question $q_j \in \mathcal{D} \setminus S_i$ that minimizes a composite score:

$$q_j = \underset{q_j \in \mathcal{D} \setminus S_i}{\arg\min} \left[ \lambda \tilde{\mathcal{A}}_{ij} + (1-\lambda) \max_{q_u \in S_i} \Sigma_{uj} \right]. \tag{5}$$

This approach not only relaxes the original NP-hard problem but also significantly improves memory efficiency by computing only $O(k \cdot |\mathcal{D}|)$ pairwise similarities on the fly. As a result, the $k$-greedy strategy is both fast and scalable, enabling efficient selection over large unannotated corpora while maintaining a strong trade-off between difficulty and diversity.

Figure 2 provides a qualitative validation of our $k$-greedy sampler's dual objectives. In the initial representation space encoded by Sentence-Transformer (left), the chosen subset spans multiple semantic regions of the full corpus, confirming that the on-the-fly diversity term successfully prevents redundant sampling. After projecting into the factorized embedding space (right), a smooth gradient of question difficulty emerges, and the selection concentrates almost in the most challenging questions. Together, these two views demonstrate that our *Difficulty–Diversity Collaborative Filtering* simultaneously maintains semantic diversity and precisely targets high-difficulty examples.

Therefore, the selected subset $S_i$ provides a highly informative slice of the large corpus for downstream use. In the case of unannotated corpora, DDCF enables cost-effective data preparation by concentrating annotation efforts, either from human experts or strong teacher models, on only the

most impactful $k$ examples. Here, DDCF serves as a front-end filter that reduces supervision costs while preserving strong learning signals. For already annotated corpora, DDCF serves as a post-hoc filter that eliminates trivial or redundant examples and tailors the learning path to the strengths and weaknesses of the target model, thereby shortening the training time. In both scenarios, the resulting compact, model-aware dataset $S_i$ can undergo further quality checks—such as expert review of selected questions and annotations— especially in high-stakes domains like medicine or law. Overall, DDCF facilitates a data-efficient tuning paradigm where LLMs can be rapidly adapted with minimal supervision, even when full-corpus annotation is impractical or prohibitively expensive.

## 4 EXPERIMENTS WITH PRE-ANNOTATED CORPUS

### 4.1 EXPERIMENTAL SETUP

**Training the Correctness Predictor**   To learn factorized model and question embeddings, we train a correctness predictor using outputs from 23 open-source LMs spanning a wide range of sizes and architectures. Each model was evaluated on the seed dataset of 19,470 questions from the training splits of GSM8K (Cobbe et al., 2021) and MATH (Hendrycks et al., 2021b). Responses were labeled correct or incorrect by a rule-based verifier, resulting in binary supervision for each model–question pair. We use $10\%$ of the questions as a held-out test set. For the initial question embeddings, we employ the sentence transformer `Qwen/Qwen3-Embedding-4B`. Appendix B lists all 23 LMs and reports the computational cost of DDCF relative to alternative data-selection baselines. Appendix C provides ablation studies on the number of participating models and the size of the seed dataset within the DDCF framework.

**Data Selection**   We conduct experiments on the `OpenR1-Math-220K` dataset (licensed under Apache 2.0), which contains 225,129 math problems annotated by `DeepSeek-R1-671B` (DeepSeek-AI et al., 2025). We select 1,000 training instances from this corpus using different selection strategies. Based on ablation results, we set the difficulty–diversity trade-off in Equation 5 to $\lambda = 0.2$ by default.

**Baselines**   We compare our approach, *DDCF*, with various baselines:

- Dummy baseline: (1) **Random** randomly samples 1,000 examples; (2) **Longest** selects the 1,000 longest instruction examples; (3) **Binary Hard** randomly samples 1,000 examples that the targeted model incorrectly answers from the seed dataset;
- Annotation dependent: (4) **Least Confidence** (Settles, 2009) measures the model's confidence as the product of probabilities of the data example. (5) **Perplexity** (Marion et al., 2023) selects examples around medium perplexity; (6) **Cartography** (Swayamdipta et al., 2020) selects easy and ambiguous examples; (7) SmallToLarge (**S2L**) (Yang et al., 2024b) samples from clusters summarizing the loss trajectory of easy-to-hard questions.
- Annotation independent: (8) **DiSF** (Fan et al., 2025) chooses samples that maximize the diversity of the question embedding space via covariance eigenvalue maximization.
- Manually selected dataset: (9) **LIMO**, 817 instructive examples from Ye et al. (2025); (10) **s1.1-1K**, 1,000 high-quality examples curated by Muennighoff et al. (2025).

It is worth noting that, unlike **Random**, **DiSF**, and our method **DDCF**—which can be applied *prior to annotation*—the remaining baselines require full-corpus annotation to compute selection criteria such as gradients, reasoning length, or perplexity. We do not compare our method with selectors like AlpaGasus (Chen et al., 2024) or LESS (Xia et al., 2024), as they assume different settings, such as reliance on ChatGPT feedback or access to a target dataset for gradient matching.

**Evaluation**   We evaluate on 10 popular reasoning benchmarks, grouped into three categories:

- **In-Distribution**: **MATH500** (Hendrycks et al., 2021b), **OlympiadBench** (He et al., 2024), **GSM8K** (Cobbe et al., 2021), **AIGEval-SAT-Math** (Zhong et al., 2024), and **AIME24**.
- **Out-of-Distribution**: **Minerva** (Lewkowycz et al., 2022), which includes undergraduate-level STEM problems; **Gaokao**, sourced from China's 2024 National College Entrance Exam; and the **STEM** subset of MMLU (Hendrycks et al., 2021a).

Table 2: Performance on In-Distribution and Out-of-Distribution benchmarks.

| Method | In-Distribution | | | | | | Out-of-Distribution | | | |
|---|---|---|---|---|---|---|---|---|---|---|
| | AIME24 | MATH | OlyBen | GSM8k | SAT | Avg. | Miverva | Gaokao | STEM | Avg. |
| **Qwen2.5-Math-7B** | | | | | | | | | | |
| Full Dataset | 64.5 | 80.6 | 42.5 | 92.6 | 98.2 | 75.8 | 46.3 | 72.2 | 79.7 | 66.1 |
| Base Model | 34.6 | 55.4 | 16.4 | 91.6 | 80.0 | 55.6 | 12.9 | 67.1 | 67.7 | 49.2 |
| Random | 38.6 | 76.4 | 34.8 | 91.0 | 98.2 | 67.8 | 41.2 | 64.6 | 75.7 | 60.5 |
| Longest | 19.7 | 53.8 | 18.4 | 81.1 | 69.1 | 48.4 | 25.7 | 36.7 | 50.7 | 37.7 |
| Binary Hard | 29.6 | 67.2 | 28.3 | 89.3 | 85.5 | 60.0 | 31.6 | 53.2 | 67.9 | 50.9 |
| Least Confid. | 12.3 | 42.8 | 11.7 | 61.6 | 58.6 | 37.4 | 21.0 | 20.3 | 47.0 | 29.4 |
| Cartography | 48.1 | 74.6 | 34.7 | 88.9 | 96.8 | 68.6 | 41.7 | 74.7 | 73.2 | 63.2 |
| Perplexity | 44.7 | 77.8 | 37.8 | 89.3 | 96.8 | 69.3 | 46.7 | 69.6 | 79.1 | 65.1 |
| S2L | 36.7 | 74.4 | 34.8 | 90.1 | 98.2 | 66.9 | 39.3 | 58.2 | 75.4 | 57.7 |
| DiSF | 44.6 | 76.2 | 35.4 | 89.9 | 96.8 | 68.6 | 43.4 | 68.4 | 75.6 | 62.4 |
| LIMO | 41.1 | 76.4 | 35.7 | 89.5 | 94.6 | 67.4 | 35.3 | 58.2 | 74.5 | 56.0 |
| s1.1-1K | 41.9 | 76.6 | 37.4 | 90.3 | 96.8 | 68.5 | 40.1 | 67.1 | 75.9 | 61.0 |
| **DDCF** | 49.0 | 77.6 | 35.0 | 91.2 | 98.2 | 70.2 | 45.6 | 74.7 | 75.8 | 65.4 |
| **Qwen3-8B-Base** | | | | | | | | | | |
| Full Dataset | 88.6 | 91.8 | 60.3 | 95.0 | 99.1 | 87.0 | 64.3 | 84.8 | 92.1 | 80.4 |
| Base Model | 47.8 | 60.8 | 36.3 | 89.8 | 98.2 | 66.6 | 40.8 | 58.2 | 84.4 | 61.1 |
| Random | 80.9 | 89.2 | 53.8 | 94.4 | 99.6 | 83.6 | 62.5 | 83.5 | 90.8 | 79.0 |
| Longest | 81.4 | 90.4 | 54.7 | 94.4 | 99.6 | 84.1 | 64.3 | 84.8 | 90.8 | 80.0 |
| Binary Hard | 75.0 | 91.4 | 54.5 | 94.2 | 93.6 | 81.8 | 60.3 | 80.0 | 86.9 | 75.7 |
| Least Confid. | 71.6 | 89.8 | 52.6 | 94.8 | 99.6 | 81.7 | 62.5 | 81.0 | 90.6 | 78.0 |
| Cartography | 78.7 | 91.6 | 56.9 | 96.7 | 99.6 | 84.7 | 63.8 | 79.9 | 91.2 | 78.3 |
| Perplexity | 79.3 | 89.8 | 55.0 | 94.5 | 99.6 | 83.6 | 60.3 | 83.5 | 91.0 | 78.3 |
| S2L | 76.4 | 91.0 | 55.0 | 94.5 | 99.1 | 83.2 | 62.1 | 78.5 | 91.3 | 77.3 |
| DiSF | 74.9 | 90.6 | 54.8 | 94.6 | 99.6 | 82.9 | 65.1 | 83.5 | 91.1 | 79.9 |
| LIMO | 79.8 | 89.4 | 55.3 | 93.7 | 98.6 | 83.4 | 54.8 | 82.3 | 88.7 | 75.2 |
| s1.1-1K | 75.5 | 86.2 | 51.9 | 92.0 | 98.2 | 80.7 | 57.7 | 77.2 | 89.3 | 74.7 |
| **DDCF** | 82.2 | 91.0 | 56.0 | 95.9 | 100 | 85.0 | 66.2 | 84.8 | 90.6 | 80.5 |

- **Development Set**: We use **SVAMP** (Patel et al., 2021) (elementary), and **AMC23** (competition level) to determine hyperparameter $\lambda$ balancing the difficulty-diversity trade-off.

We report pass@1 accuracy by default, while for **AMC23** and **AIME24** we report pass@32, due to their small size and high difficulty. Experiment details can be found in Appendix A.

## 4.2 MAIN RESULTS

**In-Distribution Results** Table 2 demonstrates that **DDCF** consistently produces the strongest 1K-example subsets among all selection strategies. For *Qwen2.5-Math-7B*, DDCF attains an average score of 70.2, outperforms the best baseline (Perplexity, 69.3) while staying within only -5.6 points of full-data training. Notably, DDCF yields larger gains on the hardest benchmarks: it boosts AIME24 performance to 49.0, +10.4 over random and +14.4 over the base model. For *Qwen3-8B-Base*, DDCF achieves 85.0 on average, outperforming all baselines and reducing the gap to full-data training to just -2.0. Its improvement is most evident on GSM8K, where DDCF reaches 95.9, surpassing all baselines by up to +3.9. These results indicate that DDCF maintains both breadth and depth in coverage, enabling efficient fine-tuning with limited data.

**Out-of-Distribution Results** Under distribution shifts, DDCF also demonstrates strong generalization. For the 7B model, it records a 65.4 average—closing the gap to the full dataset down to 0.7 and surpassing every other subset strategy by margins ranging from +0.3 to +8.0. On Gaokao, DDCF not only outperforms all baselines but also exceeds the full-data performance by +2.5 (74.7

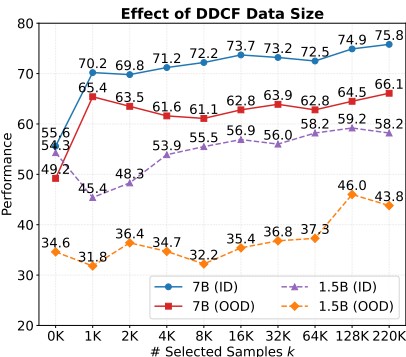
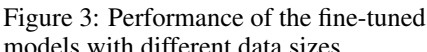
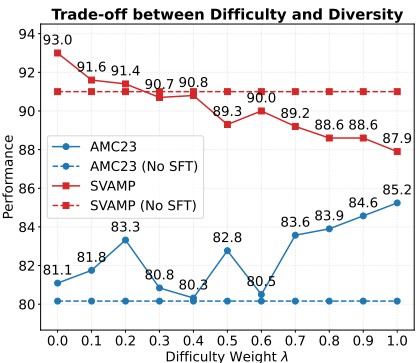

Figure 3: Performance of the fine-tuned models with different data sizes.

Figure 4: Difficulty-Diversity trade-off in data selection.

vs 72.2), suggesting that efficient fine-tuning might preserve generalization in multi-lingual settings. In the meanwhile, **Perplexity** also achieves notable results on Minerva (46.7), and STEM (79.1). For the 8B model, DDCF achieves the highest OOD average (80.5), slightly ahead of full-data fine-tuning (80.4). This edge comes primarily from Minerva, where DDCF improves by +1.9. Together, these findings highlight that compact, model-aware subsets selected by DDCF can preserve or even enhance out-of-distribution robustness relative to training on the full corpus.

Owing to space limitations, we report the results of `Falcon-10B-Base` in Appendix D.

## 4.3 DATA SIZE ABLATION

To assess the effect of training set size, we vary the number of selected questions $k$ from 0 to 225,129 and evaluate both in-distribution (ID) and out-of-distribution (OOD) performance averaged across the benchmarks introduced earlier (Figure 3). We compare two settings: a strong base model `Qwen2.5-Math-7B` and a weaker base model `Qwen2.5-Math-1.5B`.

On one hand, for `Qwen2.5-Math-7B`, ID accuracy improves almost monotonically, rising from 55.6 at $k = 0$ to 75.8 at full scale, with the sharpest gain achieved within the first 1,000 samples (70.2). OOD performance, however, exhibits a non-monotonic trend: it peaks early at 65.4 for $k = 1,000$, declines to around 61 at $k \in [4,000, 8,000]$, and then recovers steadily to 66.1 at full scale. This mid-range dip suggests that while small curated sets provide strong generalization, enlarging the pool without sufficient coverage may initially dilute transferability before larger sets restore robustness. Notably, selecting only 1,000 samples already secures over 70% of the ID improvement and nearly the full OOD benefit, highlighting the data efficiency of our DDCF framework.

On the other hand, fine-tuning on small yet highly complex datasets can degrade the performance of weaker language models—a phenomenon referred to as the *Small Model Learnability Gap* (Li et al., 2025) or *Long CoT Degradation* (Luo et al., 2025). Figure 3 illustrates this effect for `Qwen2.5-Math-1.5B` fine-tuned on DDCF subsets of size $k$. With only $k = 1,000$ examples, ID accuracy drops sharply from 54.3 to 45.4 (-8.9) and OOD falls from 34.6 to 31.8 (-2.8), illustrating the known phenomenon. Increasing to $k = 4,000$ largely mitigates this effect—ID is only 0.4 below its pre-fine-tuning level while at $k = 8,000$ both curves recover fully and begin to climb.

Beyond $k = 8,000$, performance increases steadily: by $k = 16,000$ we attain 56.9 ID and 35.4 OOD, and by $k = 128,000$ the model culminates at 59.2 ID and 46.0 OOD. Notably, this represents 1.0 ID / 2.2 OOD improvements over a conventional full-corpus fine-tuning on all 225,129 available samples, demonstrating that our DDCF strategy can overcome initial degradation and ultimately yield superior results with far fewer examples.

## 4.4 DIFFICULTY-DIVERSITY TRADE-OFF

To determine the optimal difficulty weight $\lambda$ in Equation 5, we perform an ablation study on elementary-level SVAMP and competition-level AMC23 with selection size $k = 1000$ using

`Qwen2.5-Math-7B`, as shown in Figure 4. As $\lambda$ increases from 0 (pure diversity) to 0.2, AMC23 performance jumps from 81.1 to 83.3% while SVAMP remains at its pre-trained baseline of $\sim 91\%$. Further increasing $\lambda$ continues to boost AMC23, peaking at 85.2 for $\lambda = 1.0$, but with diminishing returns, SVAMP performance declines by about 2 points at $\lambda = 0.5$ and 4 points at $\lambda = 1.0$, indicating that excessive emphasis on difficult examples undermines proficiency on simpler tasks.

Since our ultimate goal is to elicit the model's full problem-solving ability from a small, curated fine-tuning set without eroding its pre-trained knowledge, we adopt $\lambda = 0.2$ as the default parameter for our *Difficulty–Diversity Collaborative Filtering* framework, striking a balanced trade-off between difficulty and diversity. Beyond this default, DDCF enables the difficulty weight $\lambda$ to be adjusted on the fly, allowing users to instantly tailor data selection to their priorities without additional retraining or redesigning the framework. This adaptability makes the framework both convenient and versatile, supporting a wide spectrum of selection strategies within a unified formulation. Moreover, the ablation on how the quality of the large corpus affects data selection is discussed in Appendix E.

### 4.5 DOES DDCF FRAMEWORK LEARN MODEL CHARACTERISTICS?

Let $\mathcal{S}_a$ and $\mathcal{S}_b$ be subsets selected by models $\mathcal{M}_a$ and $\mathcal{M}_b$. We quantify their overlap via the Jaccard index: $J(\mathcal{S}_a, \mathcal{S}_b) = \frac{|\mathcal{S}_a \cap \mathcal{S}_b|}{|\mathcal{S}_a \cup \mathcal{S}_b|}$, which measures the fraction of questions chosen by both models relative to the total unique questions. A higher $J$ indicates greater similarity in the subsets, reflecting closer alignment in model behavior. We hypothesize that models within the same family, sharing architecture and pre-training data, will exhibit higher Jaccard similarity than those from different families. Indeed, our analysis shows an average intra-family index of **0.224** versus **0.169** inter-family, demonstrating that our framework captures meaningful model-specific characteristics.

Figure 5 shows the topic-wise composition of each model's selected subset alongside the full dataset distribution. Although the full corpus is dominated by Algebra (48.1%), our framework tailors sampling to each model's behavior. In particular, `Qwen2.5-Math-7B` and `Qwen2.5-32B` exhibit almost identical distributions: Combinatorics holds the largest share, while Algebra, Geometry, Number Theory, and Logic & Puzzles each retain substantial and balanced proportions. By contrast, `Llama-3.1-8B` diverges markedly, de-emphasizing Algebra and boosting Combinatorics and Logic & Puzzles. This divergence shows that DDCF tailors question selection to each model's specific strengths and weaknesses, targeting areas for improvement rather than sampling uniformly. The experiment on data transferability across models is in Appendix F.

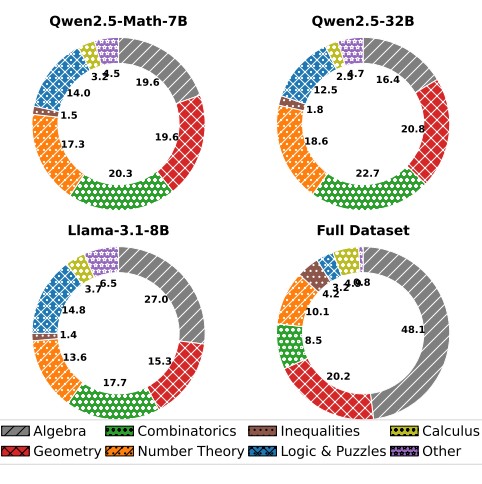

Figure 5: Topic distribution of DDCF datasets.

## 5 EXPERIMENTS WITH UNANNOTATED CORPUS AT SELECTION TIME

To illustrate capabilities of DDCF on large corpora that are initially unannotated, we evaluate on the MMLU benchmark (Hendrycks et al., 2021a), which includes 99,842 training, 1,531 validation, and 14,042 test questions. While the corpus spans many disciplines, it lacks reasoning annotations. We therefore train a *correctness predictor* on the validation split and use it with DDCF to select 1,000 high-quality training examples, which are then distilled into reasoning traces using `Qwen/Qwen3-32B` in long-thinking mode. By filtering *before* annotation, DDCF reduces distillation cost nearly $100\times$, whereas prior methods require annotating the full corpus in advance.

Table 3 compares **DDCF** with annotation-free baselines (**Random**, **DiSF**) across three MMLU domains. On *Qwen2.5-7B*, **DDCF** achieves the best accuracy in all domains, improving the average by +6.2 over the base model and +2.5 over the strongest baseline, with the largest gain in STEM (81.1; +9.6 over base, +6.4 over DiSF). For *Qwen3-8B-Base*, DDCF again excels, setting new highs in Social Science (87.3) and STEM (89.7), and raising the average by +5.2 over base with only a slight

Table 3: Performance on the MMLU benchmark.

| Method | Qwen2.5-7B | | | Qwen3-8B-Base | | | Falcon-10B-Base | | |
|---|---|---|---|---|---|---|---|---|---|
| | Humanities | Social Science | STEM | Humanities | Social Science | STEM | Humanities | Social Science | STEM |
| Base Model | 59.0 | 77.1 | 71.5 | 62.8 | 82.3 | 84.4 | 66.1 | 80.9 | 81.4 |
| Random | 61.4 | 80.5 | 79.3 | 69.0 | 86.7 | 88.8 | 69.1 | 85.5 | 85.9 |
| DiSF | 62.6 | 81.3 | 74.7 | 68.9 | 87.1 | 89.1 | 69.5 | 85.8 | 84.7 |
| **DDCF** | 63.5 | 81.4 | 81.1 | 68.2 | 87.3 | 89.7 | 69.9 | 86.0 | 87.2 |

drop in Humanities. On *Falcon-10B-Base*, DDCF outperforms all baselines, boosting Humanities, Social Science, and STEM by +3.8, +5.1, and +5.8, respectively.

Overall, these results show that DDCF strengthens not only quantitative reasoning (STEM) but also inferential reasoning (Humanities, Social Science), even with scarce annotations. Beyond in-domain performance, OOD validation (Appendix G) reveals that fine-tuning on just 1,000 distilled MMLU examples transfers effectively to general tasks such as commonsense, reading comprehension, and instruction following, where DDCF achieves the best average across all backbones.

## 6 CONCLUSION

*Difficulty–Diversity Collaborative Filtering* is a novel concept for curating small, high-quality fine-tuning subsets from large unannotated corpora by balancing question difficulty, via a learned correctness predictor, and semantic diversity in embedding space. Empirically, DDCF reduces annotation costs by $100 - 200\times$ while maintaining performance comparable to the full-data baseline, and our analysis shows it naturally tailors data selection to each model's unique strengths and weaknesses.

## ACKNOWLEDGMENTS

This research/project is supported by the National Research Foundation, Singapore under its National Large Language Models Funding Initiative (AISG Award No: AISG-NMLP-2024-005), and AI Singapore Programme (AISG Award No: AISG3-PhD-2024-08-056).

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

Table 4: Epochs and batch sizes used for supervised fine-tuning across dataset sizes.

| Hyperparameter | Dataset size $k$ | | | | | | | | |
|---|---|---|---|---|---|---|---|---|---|
| | 1K | 2K | 4K | 8K | 16K | 32K | 64K | 128K | 220K |
| Epochs | 4 | 4 | 4 | 4 | 4 | 3 | 3 | 3 | 3 |
| Batch size | 32 | 32 | 32 | 32 | 32 | 64 | 128 | 128 | 128 |

## A  EXPERIMENT DETAILS

### A.1  CORRECTNESS PREDICTORS

**MLP Block**    The question encoder $h_Q$ uses a residual multilayer perceptron (MLP) block for refinement. Given an input $E^0 \in \mathbb{R}^d$, the block is defined as

$$\text{MLPBlock}(E^0) = E^0 + u, \tag{6}$$

where

$$u = W_2\big(\text{Dropout}(\gamma(W_1 \, \text{LN}(E^0)))\big). \tag{7}$$

Here, LN denotes LayerNorm, $W_1 \in \mathbb{R}^{d \times d'}$ and $W_2 \in \mathbb{R}^{d' \times d}$ are linear layers with hidden dimension $d' = 0.1 * d$, and $\gamma$ is a ReLU activation. Dropout with rate $0.8$ is applied during training. To stabilize optimization, the final projection $W_2$ is zero-initialized, making the block behave as the identity map at initialization.

**Noise Regularization**    To reduce overfitting to the limited set of models and questions, we inject Gaussian noise into both model and question embeddings during training. For model embedding $E^0_{m_i}$ and question embedding $E^0_{q_j}$, the perturbed representations are

$$E^0_{m_i} := E^0_{m_i} + \epsilon_p, \quad E^0_{q_j} := E^0_{q_j} + \epsilon_q, \tag{8}$$

where

$$\epsilon_p, \epsilon_q \sim \mathcal{N}(0, \alpha^2 I_d), \tag{9}$$

and $\alpha$ is a scalar hyperparameter controlling the perturbation scale. This stochastic perturbation acts as embedding-level data augmentation, preventing the predictor from memorizing spurious correlations in the binary correctness matrix. During inference, noise is disabled and the raw embeddings are used.

**Training Hyperparameters**    We train the correctness predictor with the Adam optimizer (weight decay $1 \times 10^{-5}$) and a cosine learning rate schedule with a warmup ratio of $0.03$. The initial learning rate is set to $1 \times 10^{-3}$, and training runs for 30 epochs. For the *OpenR1-Math-220K* dataset, we use a batch size of $1028$ and set the regularization parameter to $\alpha = 1 \times 10^{-2}$. For the *MMLU* experiment, we use a smaller batch size of $64$ and increase the regularization strength to $\alpha = 3 \times 10^{-2}$.

### A.2  SUPERVISED FINE-TUNING

We fine-tune LLMs using the `TRL`[2] library with a maximum sequence length of 16,384 tokens. Training is performed in `bfloat16` precision with the Adam optimizer, a cosine learning rate schedule, and a warmup ratio of $0.03$. Table 4 summarizes the epoch and batch size configurations across different datasizes. Experiments in this paper can be done with 2 H100 gpus.

### A.3  DATA SELECTION PROCEDURES

**Baseline Details.**    Most baselines in our SFT experiments are described in Section 4.1, but we highlight additional implementation details here. For the **Binary Hard** baseline, we randomly sample 1,000 questions that the target model answers incorrectly from the seed datasets (GSM8K

---

[2]`https://github.com/huggingface/trl`

Table 5: Models used in the DDCF framework.

| | |
|---|---|
| deepseek-ai/deepseek-math-7b-base | Qwen/Qwen2.5-Math-1.5B |
| google/gemma-2-2b | Qwen/Qwen2.5-Math-7B |
| google/gemma-2-9b | Qwen/Qwen3-0.6B-Base |
| google/gemma-2-27b | Qwen/Qwen3-1.7B-Base |
| meta-llama/Llama-3.2-1B | Qwen/Qwen3-14B-Base |
| meta-llama/Llama-3.2-3B | Qwen/Qwen3-4B-Base |
| meta-llama/Llama-3.1-8B | Qwen/Qwen3-8B-Base |
| mistralai/Mistral-7B-v0.3 | tiiuae/Falcon3-1B-Base |
| mistralai/Mistral-Nemo-Base-2407 | tiiuae/Falcon3-3B-Base |
| Qwen/Qwen2.5-7B | tiiuae/Falcon3-7B-Base |
| Qwen/Qwen2.5-14B | tiiuae/Falcon3-10B-Base |
| Qwen/Qwen2.5-32B | |

Table 6: Runtime comparisons (using $8\times$H100 GPUs) of data selection methods.

| Method | DDCF | Perplexity | Cartography | S2L | DiSF | Random |
|---|---|---|---|---|---|---|
| Qwen2.5-Math-7B | 1.4 | 1.3 | 1.3 | 2.0 | 0.1 | 0.0 |
| Qwen3-8B-Base | 1.4 | 1.6 | 1.6 | 2.0 | 0.1 | 0.0 |
| Falcon-10B-Base | 1.4 | 2.1 | 2.1 | 2.0 | 0.1 | 0.0 |
| **Total** | 1.4 | 5.0 | 5.0 | 2.0 | 0.1 | 0.0 |

and MATH). Since GSM8K and MATH are annotated with short-CoT rationales of lower quality compared to OpenR1-Math-220K—which provides long-CoT annotations with structured reasoning and rigorous reflections—we re-annotate these 1,000 questions using `Qwen/Qwen3-32B` in long-thinking mode.

For the **S2L** method, we follow Yang et al. (2024b) and train a Pythia-70M model (Biderman et al., 2023) on the full OpenR1-Math-220K corpus as a proxy model to record loss trajectories. Samples are then clustered into 1,000 groups, from which representative examples are selected to form the training subset.

Finally, **DiSF** requires converting text samples into embedding vectors prior to selection. For a fair comparison with our proposed DDCF, we use `Qwen/Qwen3-Embedding-4B` (Zhang et al., 2025b) as the sentence encoder for DiSF.

**Data Cleaning.** Due to computational constraints, we restrict training to a maximum sequence length of 16,384 tokens. Accordingly, we discard all examples exceeding this length (fewer than 1% of OpenR1-Math-220K). To enhance the diversity of the selected dataset, we further remove duplicated questions, retaining only the instance with the shortest completion. After cleaning, the OpenR1-Math-220K dataset contains 189,257 examples, which we use for all experiments involving data selection.

### A.4 INFERENCE HYPERPARAMETERS

To improve efficiency, we accelerate inference with the SGLang framework (Zheng et al., 2024). By default, we report pass@1 accuracy, generating a single sampled response per query with `temperature=0.6`, `top_p=0.95`, `top_k=20`, `min_p=0`, and a maximum sequence length of 16,384 tokens. For the **AMC23** and **AIME24** benchmarks, we sample 64 responses per query and report pass@32 due to their small size and high difficulty.

Table 7: Effect of the number of participating models on the correctness predictor's accuracy.

| # Models | 1 | 2 | 4 | 8 | 16 | 23 |
|---|---|---|---|---|---|---|
| Accuracy | 81.5 | 81.7 | 81.8 | 82.2 | 82.5 | 82.7 |

Table 8: Effect of the number of seeding questions on the correctness predictor's accuracy.

| # Questions | 1K | 2K | 4K | 8K | 16K | 17.5K |
|---|---|---|---|---|---|---|
| Accuracy | 80.0 | 80.1 | 80.8 | 81.9 | 82.4 | 82.7 |

Table 9: Effect of Number of Participating Models on ID Performance.

| # Models | 1 | 2 | 4 | 8 | 16 | 23 |
|---|---|---|---|---|---|---|
| Accuracy | 69.5 | 70.1 | 69.9 | 69.8 | 69.9 | 70.2 |

Table 10: Effect of Number of Seeding Questions on ID Performance.

| # Questions | 1K | 2K | 4K | 8K | 16K | 17.5K |
|---|---|---|---|---|---|---|
| Accuracy | 69.3 | 68.5 | 69.5 | 70.4 | 70.0 | 70.2 |

Table 11: Effect of Number of Participating Models on OOD Performance.

| # Models | 1 | 2 | 4 | 8 | 16 | 23 |
|---|---|---|---|---|---|---|
| Accuracy | 61.7 | 60.6 | 63.5 | 62.2 | 63.8 | 65.4 |

Table 12: Effect of Number of Seeding Questions on OOD Performance.

| # Questions | 1K | 2K | 4K | 8K | 16K | 17.5K |
|---|---|---|---|---|---|---|
| Accuracy | 63.0 | 62.4 | 61.7 | 63.2 | 64.0 | 65.4 |

## B   LIST OF MODELS IN DDCF AND EFFICIENT DDCF RUNTIME

Table 5 lists the 23 models included in our DDCF framework. These are models from Qwen 2.5 (Yang et al., 2025b), Qwen 2.5 Math (Yang et al., 2024a), Qwen3 (Yang et al., 2025a), Falcon 3, Mistral (Jiang et al., 2023), Llama 3 (Grattafiori et al., 2024a), Gemma 2 (Team et al., 2024), and Deepseek Math (Shao et al., 2024).

The runtime comparison between DDCF and other selection methods on pre-annotated corpora such as OpenR1-Math-220K (excluding annotation cost, where DDCF holds a clear advantage) is reported in Table 6. DDCF requires 1.4 GPU hours total, since all participating models share the same seed-set inference (using SGLang (Zheng et al., 2024), while Correctness Predictor training and the k-greedy step incur negligible cost. In contrast, Perplexity and Cartography must process the full corpus separately for each model to compute token-level probabilities, accumulating 5.0 hours—over $3\times$ the cost of DDCF. Model-agnostic methods such as S2L and DiSF are faster but produce a single fixed subset, offering no way to tailor data to different models.

## C   HOW RELIABLE IS THE CORRECTNESS PREDICTOR?

As described in Section 4.1, we trained our Correctness Predictor on a seed dataset of triplets,(`model, question, binary correctness`), comprising 23 open-source language models and 19,470 questions, with 1,947 questions (10%) held out for testing.

### C.1   CORRECTNESS PREDICTOR'S ACCURACY DIMENSION

To evaluate Correctness Predictor's reliability, we measured the predictor's accuracy on unseen test questions for the `Qwen2.5-Math-7B` model under two conditions: (1) fixing the number of models at 23 while varying the number of training questions, and (2) fixing the number of training questions while varying the number of models. Overall, the Correctness Predictor exhibits strong sample efficiency in low-data regimes alongside steady improvements as more models or questions are added. When trained with just one model, it attains 81.5% accuracy, rising to 81.7% with two models and peaking at 82.7% when all 23 models are included (Table 7). Likewise, increasing the number of seeding questions boosts accuracy from 80.0% with 1,000 examples to 81.9% with 8,000 examples, and ultimately to 82.7% with 17,523 examples (Table 8). These results confirm that our predictor is reliable even with minimal data and scales effectively: most gains emerge early, with incremental benefits thereafter.

Table 13: Performance on In-Distribution and Out-of-Distribution benchmarks.

| Method | In-Distribution | | | | | | Out-of-Distribution | | | |
|---|---|---|---|---|---|---|---|---|---|---|
| | AIME24 | MATH | OlyBen | GSM8k | SAT | Avg. | Miverva | Gaokao | STEM | Avg. |
| **Falcon-10B-Base** | | | | | | | | | | |
| Full Dataset | 83.8 | 90.4 | 56.3 | 95.2 | 99.1 | 85.0 | 64.3 | 82.3 | 91.7 | 79.4 |
| Base Model | 41.1 | 68.6 | 34.2 | 81.4 | 93.6 | 63.8 | 39.7 | 55.7 | 81.4 | 58.9 |
| Random | 65.5 | 82.2 | 47.0 | 93.1 | 98.6 | 77.3 | 58.1 | 79.8 | 89.2 | 75.7 |
| Longest | 68.3 | 82.0 | 45.8 | 91.1 | 88.6 | 75.2 | 56.6 | 58.2 | 82.8 | 65.9 |
| Binary Hard | 67.4 | 83.4 | 49.0 | 94.2 | 77.8 | 74.3 | 57.4 | 50.6 | 76.2 | 61.4 |
| Least Confid. | 49.4 | 79.0 | 40.3 | 94.3 | 97.7 | 72.1 | 54.4 | 43.0 | 89.1 | 62.2 |
| Cartography | 63.8 | 82.2 | 43.1 | 93.8 | 97.7 | 76.1 | 59.9 | 81.1 | 89.9 | 76.9 |
| Perplexity | 60.1 | 82.8 | 45.2 | 93.5 | 99.1 | 76.1 | 61.8 | 78.5 | 89.6 | 76.6 |
| S2L | 62.2 | 82.4 | 49.2 | 94.0 | 98.6 | 77.3 | 61.0 | 74.7 | 90.0 | 75.2 |
| DiSF | 63.2 | 83.0 | 47.7 | 93.4 | 98.6 | 77.2 | 62.1 | 72.2 | 89.2 | 74.5 |
| LIMO | 66.5 | 81.4 | 48.7 | 93.5 | 57.3 | 69.5 | 51.5 | 48.1 | 68.3 | 55.9 |
| s1.1-1K | 54.8 | 80.0 | 46.7 | 93.0 | 91.8 | 73.3 | 58.1 | 65.8 | 85.8 | 69.9 |
| **DDCF** | 66.6 | 83.0 | 46.1 | 93.9 | 98.1 | 77.6 | 60.3 | 78.5 | 88.9 | 75.9 |

## C.2 DOWNSTREAM PERFORMANCE DIMENSION

We also measure the In-Distribution and Out-of-Distribution of the `Qwen2.5-Math-7B` model under the two above conditions, shown in Table 9, Table 10, Table 11, and Table12.

**ID performance is strikingly stable.** Accuracy remains tightly concentrated (68.5–70.4) across all settings, with a mean of 69.8 and a standard deviation of 0.25. This shows that DDCF reliably identifies strong in-domain training samples even with substantially fewer seed models or questions.

**OOD performance benefits from scale.** Increasing the number of participating models improves OOD accuracy from 61.7 (1 model) to 65.4 (23 models). Similarly, expanding seeding questions improves OOD accuracy from 63.0 (1K) to 65.4 (17.5K). This shows that larger seed matrices provide richer difficulty signals and improve generalization.

Overall, ID performance is robust even under very small seed budgets, while OOD performance improves steadily with more seed models and questions. This efficiency–scaling pattern makes DDCF cost-effective for in-domain fine-tuning and scalable when stronger OOD robustness is desired.

## D  EXPERIMENT RESULTS ON OPENR1-MATH-220K FOR FALCON-10B-BASE

**In-Distribution** Table 13 shows that for *Falcon-10B-Base*, **DDCF** delivers the strongest overall subset, reaching an average of 77.6. This slightly surpasses the best-performing baselines (*Random* and *S2L*, both 77.3) and narrows the gap to the full-data upper bound (85.0) to just -7.4. Performance gains are especially visible on MATH500 (83.0) and GSM8k (93.9), where DDCF matches or exceeds competing selectors. On the most challenging benchmark, AIME24, DDCF secures 66.6—well above *Perplexity* (60.1) and *Least Confidence* (49.4), underscoring its ability to capture harder examples without sacrificing breadth.

**Out-of-Distribution** On OOD tasks, DDCF remains highly competitive. It achieves an average of 75.9, ranking just behind *Perplexity* (76.6) but outperforming all other baselines, including Random (75.7) and S2L (75.2). Notably, DDCF preserves strong performance across datasets: it improves over Random on Gaokao (+-0.7 vs +13.9 over weaker baselines) and stays close to the top scorer on Minerva (60.3 vs 61.8 with Perplexity). Again, DDCF consistently produces a compact subset that balances difficulty and diversity, yielding competitive results with only 1,000 examples.

Table 14: Random selection on a synthesized "bad" corpora.

| Method | In-Distribution | | | | | | Out-of-Distribution | | | |
|---|---|---|---|---|---|---|---|---|---|---|
| | AIME24 | MATH | OlyBen | GSM8k | SAT | Avg. | Minerva | Gaokao | STEM | Avg. |
| **Qwen2.5-Math-7B** | | | | | | | | | | |
| Base Model | 34.6 | 55.4 | 16.4 | 91.6 | 80.0 | 55.6 | 12.9 | 67.1 | 67.7 | 49.2 |
| Random Low | 39.6 | 72.9 | 30.6 | 91.4 | 96.8 | 66.3 | 40.1 | 68.6 | 73.8 | 63.2 |
| Random High | 38.6 | 76.4 | 34.8 | 91.0 | 98.2 | 67.8 | 41.2 | 64.6 | 75.7 | 60.5 |
| **DDCF** | 49.0 | 77.6 | 35.0 | 91.2 | 98.2 | 70.2 | 45.6 | 74.7 | 75.8 | 65.4 |
| **Qwen3-8B-Base** | | | | | | | | | | |
| Base Model | 47.8 | 60.8 | 36.3 | 89.8 | 98.2 | 66.6 | 40.8 | 58.2 | 84.4 | 61.1 |
| Random Low | 72.1 | 89.0 | 54.8 | 96.4 | 98.6 | 82.2 | 63.0 | 80.0 | 90.2 | 77.7 |
| Random High | 80.9 | 90.4 | 53.8 | 94.4 | 99.6 | 83.6 | 62.5 | 83.5 | 90.8 | 79.0 |
| **DDCF** | 82.2 | 91.0 | 56.0 | 95.9 | 100 | 85.0 | 66.2 | 84.8 | 90.6 | 80.5 |
| **Falcon-10B-Base** | | | | | | | | | | |
| Base Model | 41.1 | 68.6 | 34.2 | 81.4 | 93.6 | 63.8 | 39.7 | 55.7 | 81.4 | 58.9 |
| Random Low | 60.2 | 81.4 | 42.1 | 93.8 | 97.3 | 75.0 | 57.2 | 72.1 | 89.6 | 73.0 |
| Random High | 65.5 | 82.2 | 47.0 | 93.1 | 98.6 | 77.3 | 58.1 | 79.8 | 89.2 | 75.7 |
| **DDCF** | 66.6 | 83.0 | 46.1 | 93.9 | 98.1 | 77.6 | 60.3 | 78.5 | 88.9 | 75.9 |

# E    IMPACT OF THE LARGE CORPUS QUALITY ON THE SUCCESS OF DATA SELECTION

The impressive performance of random selection in Table 2 and Table 13 raises the question of when it can replace more sophisticated approaches. Its success, however, is conditioned on the quality of the large corpus. If the large corpus $\mathcal{D}$ is dominated by easy or homogeneous examples, the lack of difficulty variation limits any selection strategy, and downstream performance inevitably plateaus.

To examine this effect, we construct a synthetic "bad" corpus $\mathcal{D}^{\text{syn}}$ composed primarily of easy questions. From the remaining pool, we sample 19,000 "bad" items (maximizing the composite score in Equation 5) and 1,000 "cherry" items (minimizing the score), forming a highly skewed dataset where only 5% of random samples are genuinely high-quality. The results across three backbones are reported in Table 14.

Across Qwen2.5-Math-7B, Qwen3-8B-Base, and Falcon-10B-Base, **Random Low** (sampling from $\mathcal{D}^{\text{syn}}$) consistently underperforms **Random High** (sampling from the full OpenR1-Math-220K) on challenging benchmarks such as AIME24, MATH, and OlympiadBench. The gaps are substantial—for instance, $-3.3$ (MATH) and $-4.2$ (OlympiadBench) on Qwen2.5-Math-7B, $-8.2$ (AIME24) on Qwen3-8B-Base, and $-5.3$ (AIME24) on Falcon-10B-Base.

These results indicate that while LLMs can learn easy tasks without curated data, high-quality and diverse examples remain crucial for strong performance on difficult reasoning benchmarks. Random sampling from an easy-heavy corpus cannot approach expert-level performance, e.g., AIME24.

# F    DATA TRANSFERABILITY BETWEEN MODELS

Figure 6 shows the performance of `Qwen2.5-Math-7B` after fine-tuning on DDCF datasets curated for other models. Fine-tuning on its own curated data yields the highest combined performance of 67.5%. Substituting the dataset from `Qwen2.5-32B` incurs a modest 0.4 point drop (to 67.1%), while using `Gemma-2-9B` and `Mistral-7B-v0.3` subsets leads to declines of 2.0 and 2.3 points, respectively. Beyond these, we observe a gradually larger drop of 2.5 points with `Qwen2.5-Math-1.5B` and `Falcon3-7B-Base`, and 2.7 points with `Llama-3.1-8B`. Overall, this pattern hints that datasets drawn from models with closer architectural or training kinship may transfer more effectively, although more extensive experiments would be needed to confirm the precise nature of this relationship.

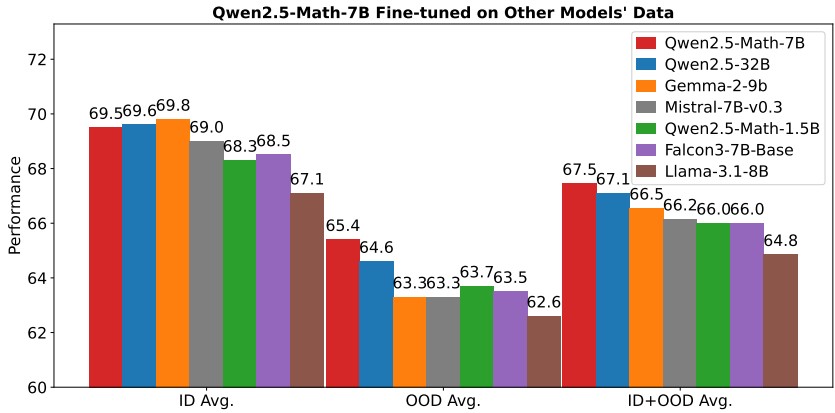

Figure 6: Each model has its own datasets. Using other models' datasets yields suboptimal results.

# G  OOD PERFORMANCE ON GENERAL TASKS OF LLMS FINE-TUNED ON MMLU SUBSETS

Table 15: Performance on OOD general tasks.

| Method | Qwen2.5-7B | | | Qwen3-8B-Base | | | Falcon-10B-Base | | | Avg. |
|---|---|---|---|---|---|---|---|---|---|---|
| | LogiQA | OpenBookQA | AlpacaEval2.0 | LogiQA | OpenBookQA | AlpacaEval2.0 | LogiQA | OpenBookQA | AlpacaEval2.0 | |
| Base Model | 47.3 | 83.6 | 5.6 | 51.8 | 82.6 | 16.5 | 48.1 | 80.8 | 7.0 | 47.0 |
| Random | **50.7** | 89.4 | 33.3 | 61.0 | 93.2 | 59.3 | 53.5 | 90.4 | 49.7 | 64.5 |
| DiSF | 47.0 | 88.4 | **36.6** | 60.8 | **94.4** | 58.3 | 52.8 | 90.0 | 46.0 | 63.8 |
| **DDCF** | 48.2 | **90.4** | 32.5 | **61.3** | 92.0 | **59.5** | **56.2** | **92.0** | **53.3** | **65.0** |

While DDCF is tailored for reasoning-centric MMLU tasks, Table 15 shows it also transfers effectively to out-of-distribution (OOD) general tasks. Fine-tuning on just 1,000 distilled MMLU examples leads to strong performance across diverse benchmarks, including commonsense reasoning (LogiQA (Liu et al., 2020)), reading comprehension (OpenBookQA (Mihaylov et al., 2018)), and instruction following (AlpacaEval 2.0 (Dubois et al., 2024)), without using any target-task labels.

DDCF outperforms the base models by an average of +18.0 points and achieves the highest overall average (65.0) among all methods. On average, it improves commonsense reasoning by +6.2 over Base, delivers state-of-the-art reading comprehension on Qwen2.5 and Falcon (+9.1 avg), and shows the largest gains in instruction following (+38.7), surpassing Random and DiSF on stronger backbones. These results underscore DDCF's broad generalization ability beyond its intended domain.

