# OpenReview forum: "Difficulty–Diversity Collaborative Filtering for Data-Efficient LLM Fine-Tuning"
_ICLR.cc/2026/Conference — ICLR 2026 Poster_

### Official Review · Reviewer_b2jG · 2025-10-16

**Soundness:** 2
**Presentation:** 3
**Contribution:** 2
**Rating:** 6
**Confidence:** 4

**Summary:**

The paper introduces a new strategy for selecting a set of high-quality samples that can be used to improve fine-tuning of LLMs on different tasks. The method is designed to work with unlabelled and limited data by first training a simple linear model for predicting the difficulty of the samples based on the behaviour of different LLMs on the questions. Afterwards, the difficulty is balanced with diversity of the samples to select the set of samples. Through extensive experiments the paper shows such selection leads to set of samples that can result in better accuracy after fine-tuning.

**Strengths:**

The proposed method is simple, easy to understand and builds on the previous advances or other methods that were shown to work well. As such, it is not overly complicated and is easy to use which may allow its widespread use.

The method is evaluated through multiple experiments, ablation studies and multiple datasets and models.

Writing is very clean and paper is easy to read and understand.

**Weaknesses:**

**Missing related approaches for sample selection**

Although the paper compares with multiple baselines, there are 2 approaches that are very similar to the DDCF method, which are completely missing from the paper. This includes the "Datamodels" selection [1, 2], which also trains a linear model to predict the benefit of each sample to the model (where the DDCF can be viewed as an extension of this approach that utilises multiple models at the same time when calculating the difficulty); and "Dataset Cartography" [3, 4] which determines the difficulty of the samples based on their training dynamics and shows that combination of easy and ambiguous samples provides benefit for training. In both cases, these methods are more related to determining the difficulty of the samples and so the comparison with them would strengthen the paper -- also including them in the related work section. I acknowledge that they are mostly used for selecting in-context examples, but they are not constrained by the typical approaches there (that select for specific test sample) and so could be used here, especially considering how similar they are to the proposed method.

There are also other methods for selecting in-context examples that balance multiple properties (including diversity) that could be included as part of the related work [5, 6, 7, 8].

**Missing cost comparison across different approaches**

Based on the experimental setup, for the DDCF method to work, you run ~19.5k inferences on using 23 different models. One of the main claims of the paper is that this method is less expensive than other approaches. As such, I would expect there to be a comparison in cost with different methods across different settings -- for example, if I am interested in selecting for samples for one specific model, running the extensive inferences from the DDCF method may not be the best solution. I would suggest adding a more in-depth discussion for the cost-performance trade-off. I acknowledge that there in Appendix C, there is an analysis how the DDCF difficulty evaluation performs when changing the number of models and questions that are requires, but I would expect to evaluate this not only based on the "correctness" of the difficulty predictor but based on the samples chosen in these data-contrained settings instead.

Similarly, based on the results, the random selection achieves strong accuracy already -- it is close to the proposed DDCF method and the best-performing approaches. I would suggest adding a discussion on this in regards to the trade-offs, as random selection is quite cheaper (and does not require labels). It might be an effect of the data, i.e., some datasets that have high-quality samples already do not benefit as much from more sophisticated selection, while on others (like AIME24) that may be more complicated or noisy, it might produce significantly better results. As such, I would suggest doing a more detailed analysis for these characteristics.

A little side note: I would also suggest better references to the Appendix B and C that to a certain extent deal with this problem, but it is not as obvious from the text that it is included.


**Results presented withouth consideration for LLM randomness**

All the results in the paper are presented from a single run, although the LLMs (and fine-tuning) are affected by different effects of randomness. This affects both the "corectness predictor", where repeated fine-tuning could lead to different results, but also the LLM fine-tuning on the selected samples. Although the expected differences might be small, the differences in results across different approaches are not that high so it would be interesting to so how the method is affected by randomness.


**(minor) Writing changes**

In Table 2 (and also other tables) some numbers are bolded or underlines, but there is no explanation why. It is also done only for specific datasets. What does this mean? I would expect either bolding the best performing models (and underlining the runner-up) or removing it altogether.

There are cases on data where the DDCF method does not outperform other methods, although you claim it happends -- e.g., Qwen-2.5 on Minerva and STEM datasets is outperformed by Perplexity and/or s1.1-1K; or Qwen-3 on STEM for multiple approaches. I would suggest updating lines 356-359 as it is a bit misleading in terms of results (and not really necessary as I would not expect the method to be the best across all settings); same with Appendix D

**References**
1. Datamodels: Predicting Predictions from Training Data
2. Data Curation Alone Can Stabilize In-context Learning
3. Dataset Cartography: Mapping and Diagnosing Datasets with Training Dynamics
4. Cartography Active Learning
5. Finding Support Examples for In-Context Learning
6. Automatic Combination of Sample Selection Strategies for Few-Shot Learning
7. EXPLORA: Efficient Exemplar Subset Selection for Complex Reasoning
8. Sample Efficient Demonstration Selection for In-Context Learning

**Questions:**

See weaknesses for more details:

How does the proposed DDCF method compared with different baslines for selecting samples in terms of performance-cost trade-off (including random selection)?

What is the impact of dataset quality on the sucess of sample selection?

---

> ### Author Response · Authors · 2025-11-21
> **Response to Reviewer ksTm (Part 1/4)**
>
> Dear Reviewer b2jG,
>
> Thank you for the thoughtful reviews and constructive feedback. We sincerely appreciate the time and effort you devoted to evaluating our work. We address your points below.
>
> > **Weakness 1:** Missing related approaches for sample selection
>
> For clarity, we use "correctness predictor / datamodel" to refer to meta-models that learn properties of a target model, and target models to denote the LLMs fine-tuned on selected subsets.
>
> **Relation to Datamodels [1, 2].**
>
> Datamodels [1] learn a mapping $\\{0,1\\}^{|\mathcal{D}|} \xrightarrow{} \mathbb{R}$ to estimate how the presence or absence of each training example affects the final trained model. This differs from DDCF in several fundamental aspects:
>
> - **Timing and cost.** DDCF learns a correctness predictor before any training, using only a small seed dataset. In contrast, [1] repeatedly trains the target model on many random subsets to compute convergent datamodels—feasible only for small models (e.g., ResNet-9 with ~5M parameters) and impractical for multi-billion-parameter LLMs.
> - **Complexity and generalization.** Datamodels operate in the combinatorial space $\\{0,1\\}^{|\mathcal{D}|}$ ignoring example content. This space grows exponentially and does not generalize to unseen questions. DDCF instead uses learned content-based representations and scales to large corpora (220K examples).
> - **Diversity.** Datamodels do not model diversity, whereas DDCF jointly optimizes difficulty and diversity.
>
> Despite these differences, datamodels are conceptually complementary: they support the intuition that meta-models can approximate model behavior and that small subsets (<1%) may significantly influence performance—reinforcing the motivation for DDCF.
>
> Datamodels for few-shot selection [2] inherit similar limitations: they rely on the same $\\{0,1\\}^{|\mathcal{D}|}$ formulation, are restricted to very small datasets (1,000 examples), and cannot generalize beyond the training domain.
>
> **Relation to Dataset Cartography [3, 4].**
> *Dataset Cartography* uses token-probability dynamics to identify "easy" and "ambiguous" samples, inheriting the same limitations as perplexity-based scoring (discussed in Table 1 of the main paper). Below we compare Cartography and DDCF under identical random seeds. DDCF consistently outperforms Cartography on ID accuracy across all three models and improves OOD accuracy on two of them, with the remaining case showing a comparable ID-OOD trade-off.
>
> ---
> **Rebuttal Table 1:** Performance on In-Distribution and Out-of-Distribution benchmarks.
>
> **Qwen2.5-Math-7B**
> | **Method**  | AIME24 | MATH | OlyBen | GSM8k |  SAT | **ID Avg.** | Minerva | Gaokao | STEM | **OOD Avg.** |
> | ----------- | -----: | ---: | -----: | ----: | ---: | ----------: | ------: | -----: | ---: | -----------: |
> | Cartography |   48.1 | 74.6 |   34.7 |  88.9 | 96.8 |        68.6 |    41.7 |   74.7 | 73.2 |         63.2 |
> | **DDCF**    |   49.0 | 77.6 |   35.0 |  91.2 | 98.2 |    **70.2** |    45.6 |   74.7 | 75.8 |     **65.4** |
>
>
> **Qwen3-8B-Base**
> | **Method**  | AIME24 | MATH | OlyBen | GSM8k |  SAT | **ID Avg.** | Minerva | Gaokao | STEM | **OOD Avg.** |
> | ----------- | -----: | ---: | -----: | ----: | ---: | ----------: | ------: | -----: | ---: | -----------: |
> | Cartography |   78.7 | 91.6 |   56.9 |  96.7 | 99.6 |        84.7 |    63.8 |   79.9 | 91.2 |         78.3 |
> | **DDCF**    |   82.2 | 91.0 |   56.0 |  95.9 |  100 |    **85.0** |    66.2 |   84.8 | 90.6 |     **80.5** |
>
> **Falcon-10B-Base**
> | **Method**  | AIME24 | MATH | OlyBen | GSM8k |  SAT | **ID Avg.** | Minerva | Gaokao | STEM | **OOD Avg.** |
> | ----------- | -----: | ---: | -----: | ----: | ---: | ----------: | ------: | -----: | ---: | -----------: |
> | Cartography |   63.8 | 82.2 |   43.1 |  93.8 | 97.7 |        76.1 |    59.9 |   81.1 | 89.9 |     **76.9** |
> | **DDCF**    |   66.6 | 83.0 |   46.1 |  93.9 | 98.1 |    **77.6** |    60.3 |   78.5 | 88.9 |         75.9 |
>
> **Relation to multi-property ICL selection [5–8].**
> Given the convenience of DDCF and its potential for widespread use, the framework can readily extend beyond supervised fine-tuning to settings such as In-Context Learning [2,5,6,7,8], Active Learning [4], and even Curriculum Learning [9]. We agree that connecting DDCF to these lines of work would strengthen the paper’s positioning and will incorporate these discussions in the next version of the paper.
>
> *Note: References list is provided in the response to Part 4/4.*

---

> > ### Author Response · Authors · 2025-11-21
> > **Response to Reviewer ksTm (Part 2/4)**
> >
> > > **Weakness 3:**  Results considered LLM randomness
> >
> > To account for randomness we additionally report results averaged over 5 independent runs on OpenR1-Math-220K, despite the limited computational budget and the large number of experiments in this work. This evaluation compares three representative methods: **DDCF**, the newly added baseline Dataset **Cartography**, and the strong baseline **Perplexity**.
> >
> > ---
> > **Rebuttal Table 2:** Performance on ID and OOD benchmarks.
> >
> > **Qwen2.5-Math-7B**
> > | **Method**  | AIME24          | MATH            | OlyBen          | GSM8k           | SAT             | **ID Avg.**              | Minerva         | Gaokao          | STEM            | **OOD Avg.**             |
> > | ----------- | --------------- | --------------- | --------------- | --------------- | --------------- | ------------------------ | --------------- | --------------- | --------------- | ------------------------ |
> > | Cartography | $50.7_{\pm1.8}$ | $76.0_{\pm1.2}$ | $33.5_{\pm1.0}$ | $89.1_{\pm0.2}$ | $96.4_{\pm0.5}$ | $69.1_{\pm0.3}$          | $41.3_{\pm1.5}$ | $71.7_{\pm4.5}$ | $72.8_{\pm0.5}$ | $62.0_{\pm1.6}$          |
> > | Perplexity  | $45.6_{\pm2.4}$ | $76.1_{\pm1.5}$ | $33.1_{\pm2.5}$ | $90.7_{\pm0.8}$ | $97.0_{\pm0.6}$ | $68.5_{\pm0.8}$          | $44.7_{\pm1.4}$ | $71.2_{\pm2.4}$ | $76.7_{\pm1.6}$ | $64.2_{\pm1.1}$          |
> > | **DDCF**    | $51.6_{\pm2.0}$ | $77.6_{\pm0.5}$ | $34.5_{\pm1.0}$ | $91.5_{\pm0.6}$ | $97.4_{\pm0.5}$ | $\mathbf{70.5}_{\pm0.4}$ | $44.1_{\pm2.7}$ | $73.9_{\pm2.8}$ | $75.1_{\pm0.4}$ | $\mathbf{64.4}_{\pm1.1}$ |
> >
> > **Qwen3-8B-Base**
> > | **Method**  | AIME24          | MATH            | OlyBen          | GSM8k           | SAT             | **ID Avg.**              | Minerva         | Gaokao          | STEM            | **OOD Avg.**             |
> > | ----------- | --------------- | --------------- | --------------- | --------------- | --------------- | ------------------------ | --------------- | --------------- | --------------- | ------------------------ |
> > | Cartography | $79.7_{\pm3.0}$ | $90.8_{\pm1.0}$ | $57.3_{\pm0.4}$ | $96.7_{\pm0.4}$ | $99.5_{\pm0.5}$ | $84.8_{\pm0.5}$          | $65.6_{\pm1.2}$ | $79.9_{\pm2.3}$ | $91.1_{\pm0.3}$ | $78.9_{\pm1.0}$          |
> > | Perplexity  | $82.2_{\pm4.6}$ | $90.4_{\pm1.2}$ | $56.5_{\pm1.0}$ | $95.6_{\pm0.7}$ | $99.5_{\pm0.4}$ | $84.9_{\pm1.1}$          | $64.0_{\pm2.2}$ | $80.1_{\pm1.9}$ | $90.8_{\pm0.2}$ | $78.3_{\pm0.3}$          |
> > | **DDCF**    | $83.0_{\pm2.2}$ | $91.9_{\pm0.8}$ | $57.7_{\pm1.3}$ | $96.2_{\pm0.2}$ | $99.6_{\pm0.2}$ | $\mathbf{85.7}_{\pm0.6}$ | $66.8_{\pm0.9}$ | $81.6_{\pm2.5}$ | $90.8_{\pm0.5}$ | $\mathbf{79.7}_{\pm1.0}$ |
> >
> > **Falcon-10B-Base**
> > | **Method**  | AIME24          | MATH            | OlyBen          | GSM8k           | SAT             | **ID Avg.**              | Minerva         | Gaokao          | STEM            | **OOD Avg.**             |
> > | ----------- | --------------- | --------------- | --------------- | --------------- | --------------- | ------------------------ | --------------- | --------------- | --------------- | ------------------------ |
> > | Cartography | $65.4_{\pm3.8}$ | $82.7_{\pm0.4}$ | $43.1_{\pm0.1}$ | $93.8_{\pm0.2}$ | $97.2_{\pm0.5}$ | $76.4_{\pm0.8}$          | $60.7_{\pm2.8}$ | $77.6_{\pm3.2}$ | $90.0_{\pm0.2}$ | $\mathbf{76.1}_{\pm1.5}$ |
> > | Perplexity  | $65.4_{\pm3.6}$ | $84.4_{\pm1.0}$ | $44.6_{\pm0.6}$ | $94.0_{\pm0.4}$ | $98.0_{\pm0.7}$ | $77.2_{\pm0.9}$          | $60.3_{\pm1.6}$ | $78.1_{\pm2.8}$ | $89.2_{\pm0.3}$ | $75.9_{\pm1.0}$          |
> > | **DDCF**    | $66.8_{\pm0.5}$ | $83.6_{\pm1.1}$ | $45.4_{\pm0.6}$ | $94.0_{\pm0.2}$ | $97.7_{\pm0.5}$ | $\mathbf{77.5}_{\pm0.3}$ | $59.8_{\pm0.4}$ | $78.1_{\pm0.7}$ | $89.0_{\pm0.2}$ | $75.6_{\pm1.0}$          |
> >
> > Across all three backbones, DDCF achieves the strongest ID performance under multi-run evaluation, outperforming Cartography and Perplexity on Qwen2.5-Math-7B (70.5 vs. 69.1/68.5), Qwen3-8B-Base (85.7 vs. 84.8/84.9), and Falcon-10B-Base (77.5 vs. 76.4/77.2). DDCF also shows strong OOD generalization, leading on Qwen2.5-Math-7B (64.4) and Qwen3-8B-Base (79.7), and remaining competitive on Falcon-10B-Base (75.6 vs. 76.1), offering the best overall ID–OOD balance.
> >
> > > **Weakness 4:** (minor) Writing changes
> >
> > Thank you for pointing this out. We will clarify in the revised version that boldface indicates the best performance and underlining indicates the runner-up in all tables, ensuring consistency and avoiding ambiguity.
> >
> > Regarding the OOD results, we agree with the reviewer’s observation: DDCF does not achieve the top score on every dataset. In some settings—such as Minerva or STEM for Qwen2.5/Qwen3—methods like Perplexity or s1.1-1K perform slightly better. Our intention was not to claim universal dominance, but to show that DDCF remains reliably strong and consistently competitive across a wide range of evaluation conditions.
> >
> > We will revise the discussion around lines 356–359 (and Appendix D) to present this point more clearly.

---

> > > ### Author Response · Authors · 2025-11-21
> > > **Response to Reviewer ksTm (Part 3/4)**
> > >
> > > > **Question 1.1:** Performance-Cost Trade-off of DDCF.
> > >
> > > ---
> > > **Rebuttal Table 3:** Effect of Number of Participating Models on ID Performance
> > >
> > > | **# Models** | 1    | 2    | 4    | 8    | 16   | 23   |
> > > | ------------ | ---- | ---- | ---- | ---- | ---- | ---- |
> > > | **Accuracy** | 69.5 | 70.1 | 69.9 | 69.8 | 69.9 | 70.2 |
> > > ---
> > > **Rebuttal Table 4:** Effect of Number of Seeding Questions on ID Performance
> > >
> > > | **# Questions** | 1K   | 2K   | 4K   | 8K   | 16K  | 17.5K |
> > > | --------------- | ---- | ---- | ---- | ---- | ---- | ----- |
> > > | **Accuracy**    | 69.3 | 68.5 | 69.5 | 70.4 | 70.0 | 70.2  |
> > > ---
> > > **Rebuttal Table 5:** Effect of Number of Participating Models on OOD Performance
> > >
> > > | **# Models** | 1    | 2    | 4    | 8    | 16   | 23   |
> > > | ------------ | ---- | ---- | ---- | ---- | ---- | ---- |
> > > | **Accuracy** | 61.7 | 60.6 | 63.5 | 62.2 | 63.8 | 65.4 |
> > > ---
> > > **Rebuttal Table 6:** Effect of Number of Seeding Questions on OOD Performance
> > >
> > > | **# Questions** | 1K   | 2K   | 4K   | 8K   | 16K  | 17.5K |
> > > | --------------- | ---- | ---- | ---- | ---- | ---- | ----- |
> > > | **Accuracy**    | 63.0 | 62.4 | 61.7 | 63.2 | 64.0 | 65.4  |
> > >
> > > Thank you for the suggestion on the ablation studies. We further vary the seed matrix along two axes—(1) fixing 17.5K questions while changing the number of seed models, and (2) fixing 23 models while changing the number of seeding questions—and fine-tune Qwen2.5-Math-7B on the resulting DDCF-selected subsets (Rebuttal Tables 1–4).
> > >
> > > - **ID performance is strikingly stable.** Accuracy remains tightly concentrated (68.5–70.4) across all settings, with a mean of 69.8 and a standard deviation of 0.25. This shows that DDCF reliably identifies strong in-domain training samples even with substantially fewer seed models or questions.
> > >
> > > -  **OOD performance benefits from scale.** Increasing the number of participating models improves OOD accuracy from 61.7 (1 model) to 65.4 (23 models). Similarly, expanding seeding questions improves OOD accuracy from 63.0 (1K) to 65.4 (17.5K). This shows that larger seed matrices provide richer difficulty signals and improve generalization..
> > >
> > > -  **Summary.** ID performance is robust even under very small seed budgets, while OOD performance improves steadily with more seed models and questions. This efficiency–scaling pattern makes DDCF cost-effective for in-domain fine-tuning and scalable when stronger OOD robustness is desired.
> > >
> > > ​We will add this ablation study into the next version of the paper to comprehend the proposed DDCF.
> > >
> > > > **Question 1.2 and Weakness 2:** Cost comparison across different approaches.
> > >
> > > To isolate the overhead of each data-selection method, we compare runtime only on pre-annotated corpora such as OpenR1-Math-220K (ignoring annotation cost, where DDCF has a clear advantage). Using 8×H100 GPUs, the runtimes are:
> > >
> > > ---
> > > **Rebuttal Table 7**: Runtime comparisions (using 8xH100 GPUs) of data selection methods.
> > > | **Method**      | **DDCF** | **Perplexity** | **Cartography** | **S2L** | **DiSF** | **Random** |
> > > | --------------- | -------: | -------------: | --------------: | ------: | -------: | ---------: |
> > > | Qwen2.5-Math-7B |      1.4 |            1.3 |             1.3 |     2.0 |      0.1 |        0.0 |
> > > | Qwen3-8B-Base   |      1.4 |            1.6 |             1.6 |     2.0 |      0.1 |        0.0 |
> > > | Falcon-10B-Base |      1.4 |            2.1 |             2.1 |     2.0 |      0.1 |        0.0 |
> > > | **Total**       |  **1.4** |        5.0     |         5.0     |     2.0 |      0.1 |        0.0 |
> > >
> > >
> > > DDCF requires 1.4 GPU hours total, since all participating models share the same seed-set inference; while Correctness Predictor training and the k-greedy step incur negligible cost. In contrast, Perplexity and Cartography must process the full corpus separately for each model to compute token-level probabilities, accumulating 5.0 hours—over 3x the cost of DDCF. Model-agnostic methods such as S2L and DiSF are faster but produce a single fixed subset, offering no way to tailor data to different models.
> > >
> > > Finally, Appendix C shows that DDCF remains strong even when using far fewer seed questions or models, confirming that its seed-inference cost is bounded, scalable, and efficient across target LLMs.

---

> > > > ### Author Response · Authors · 2025-11-21
> > > > **Response to Reviewer ksTm (Part 4/4)**
> > > >
> > > > > **Question 2:** What is the impact of dataset quality on the sucess of sample selection?
> > > >
> > > > The effectiveness of any data-selection method fundamentally depends on the quality and diversity of the candidate corpus. If the corpus $\mathcal{D}$ lacks meaningful variation in example difficulty—e.g., dominated by overly easy or homogeneous problems—then no selection algorithm can extract a high-value subset, and the downstream performance will inevitably plateau.
> > > >
> > > > To illustrate this, we construct a synthetic “bad” corpus $\mathcal{D}^{\text{syn}}$ containing mostly easy and homogeneous questions. From the remaining pool, we intentionally sample 19,000 "bad" items (maximizing the composite score of Equation 5 in our paper) and 1,000 “cherry” items (minimizing the score of Equation 5 in our paper), producing a highly skewed dataset where only 5% of random samples would be truly high-quality. The evaluation results are shown below.
> > > >
> > > > ---
> > > > **Rebuttal Table 8:** Performance on In-Distribution and Out-of-Distribution benchmarks.
> > > >
> > > > **Qwen2.5-Math-7B**
> > > > | **Method**  | AIME24 | MATH | OlyBen | GSM8k |  SAT | **ID Avg.** | Minerva | Gaokao | STEM | **OOD Avg.** |
> > > > | ----------- | -----: | ---: | -----: | ----: | ---: | ----------: | ------: | -----: | ---: | -----------: |
> > > > | Random_Low  |   39.6 | 72.9 |   30.6 |  91.4 | 96.8 |        66.3 |    40.1 |   68.6 | 73.8 |         63.2 |
> > > > | Random_High |   38.6 | 76.4 |   34.8 |  91.0 | 98.2 |        67.8 |    41.2 |   64.6 | 75.7 |         60.5 |
> > > > | **DDCF**    |   49.0 | 77.6 |   35.0 |  91.2 | 98.2 |    **70.2** |    45.6 |   74.7 | 75.8 |     **65.4** |
> > > >
> > > >
> > > > **Qwen3-8B-Base**
> > > > | **Method**  | AIME24 | MATH | OlyBen | GSM8k |  SAT | **ID Avg.** | Minerva | Gaokao | STEM | **OOD Avg.** |
> > > > | ----------- | -----: | ---: | -----: | ----: | ---: | ----------: | ------: | -----: | ---: | -----------: |
> > > > | Random_Low  |   72.1 | 89.0 |   54.8 |  96.4 | 98.6 |        82.2 |    63.0 |   80.0 | 90.2 |         77.7 |
> > > > | Random_High |   80.9 | 90.4 |   53.8 |  94.4 | 99.6 |        83.6 |    62.5 |   83.5 | 90.8 |         79.0 |
> > > > | **DDCF**    |   82.2 | 91.0 |   56.0 |  95.9 |  100 |    **85.0** |    66.2 |   84.8 | 90.6 |     **80.5** |
> > > >
> > > >
> > > > **Falcon-10B-Base**
> > > > | **Method**  | AIME24 | MATH | OlyBen | GSM8k |  SAT | **ID Avg.** | Minerva | Gaokao | STEM | **OOD Avg.** |
> > > > | ----------- | -----: | ---: | -----: | ----: | ---: | ----------: | ------: | -----: | ---: | -----------: |
> > > > | Random_Low  |   60.2 | 81.4 |   42.1 |  93.8 | 97.3 |        75.0 |    57.2 |   72.1 | 89.6 |         73.0 |
> > > > | Random_High |   65.5 | 82.2 |   47.0 |  93.1 | 98.6 |        77.3 |    58.1 |   79.8 | 89.2 |         75.7 |
> > > > | **DDCF**    |   66.6 | 83.0 |   46.1 |  93.9 | 98.1 |    **77.6** |    60.3 |   78.5 | 88.9 |     **75.9** |
> > > >
> > > > Across Qwen2.5-Math-7B, Qwen3-8B-Base, and Falcon-10B-Base, **Random_Low** (random selection in bad corpus $\mathcal{D}^{\text{syn}}$) significantly performs worse than **Random_High** (Random in full corpus $\mathcal{D}$ OpenR1-Math-220K) on challenging benchmarks such as AIME, MATH, and OlympiadBench—e.g., gaps of −3.3 (MATH) and −4.2 (OlyBen) for Qwen2.5-Math-7B, −8.2 (AIME24) for Qwen3-8B-Base, and −5.3 (AIME24) for Falcon-10B-Base. Therefore, while LLMs can learn easy tasks without curated data, sophisticated selection methods are essential for challenging problems. Random sampling from an easy-heavy corpus cannot yield expert-level performance (e.g., AIME24), even with a large model.
> > > >
> > > > Finally, this empirical observation aligns with recent theory [10], which proves that under a strong base model and a corpus rich in hard examples, "keep-hard" selection is provably advantageous—supported by their experiments showing substantial gains over random sampling.
> > > >
> > > > ### **References**
> > > > 1. Ilyas, Andrew, et al. "Datamodels: Predicting predictions from training data." ICML (2022).
> > > > 2. Chang, Ting-Yun, and Robin Jia. "Data curation alone can stabilize in-context learning." ACL, 2023.
> > > > 3.  Swayamdipta, Swabha, et al. "Dataset cartography: Mapping and diagnosing datasets with training dynamics." arXiv preprint arXiv:2009.10795 (2020).
> > > > 4.  Zhang, Mike, and Barbara Plank. "Cartography active learning." arXiv preprint arXiv:2109.04282 (2021).
> > > > 5. Li, Xiaonan, and Xipeng Qiu. "Finding support examples for in-context learning." arXiv preprint arXiv:2302.13539 (2023).
> > > > 6. Pecher, Branislav, et al. "Automatic combination of sample selection strategies for few-shot learning." arXiv preprint arXiv:2402.03038 (2024).
> > > > 7. Purohit, Kiran, et al. "EXPLORA: Efficient exemplar subset selection for complex reasoning." EMNLP, 2024.
> > > > 8. Purohit, Kiran, et al. "Sample Efficient Demonstration Selection for In-Context Learning." arXiv preprint arXiv:2506.08607 (2025).
> > > > 9.  Soviany, Petru, et al, "Curriculum learning: A survey", IJCV, 2021
> > > > 10. Dohmatob, Elvis, Mohammad Pezeshki, and Reyhane Askari-Hemmat. "Why Less is More (Sometimes): A Theory of Data Curation." arXiv preprint arXiv:2511.03492 (2025).

---

> > > > > ### Comment · Reviewer_b2jG · 2025-11-21
> > > > >
> > > > > Thank you for the extensive and detailed answers. They have addressed all of my concerns and updating the paper with these additional ablations and details will surely strengthen the paper, its contribution and allow for interesting discussions at the conference. As a result, I am increasing my score.

---

> > > > > > ### Author Response · Authors · 2025-11-21
> > > > > > **Official Comment by Authors**
> > > > > >
> > > > > > Dear Reviewer b2jG,
> > > > > >
> > > > > > We sincerely appreciate your detailed review and the helpful suggestions you have provided. Thank you for the time and effort you dedicated to evaluating our work.
> > > > > >
> > > > > > We will upload the revised version once all reviewer discussions are finalized.
> > > > > >
> > > > > > Best regards,
> > > > > >
> > > > > > The Authors

---

### Official Review · Reviewer_ksTm · 2025-10-27

**Soundness:** 3
**Presentation:** 4
**Contribution:** 2
**Rating:** 4
**Confidence:** 4

**Summary:**

In this paper, the authors addressed the inefficiency of data selection from large scale datasets for LLM supervised fine-tuning: high annotation costs, catastrophic forgetting, and redundant examples, by proposing Difficulty-Diversity Collaborative Filtering (DDCF). DDCF is a data selection framework that curates compact, high-quality subsets from large (often unannotated corpora), it leverages collaborative filtering to balance two criteria in data selection: question difficulty and semantic diversity, to enable SFT with minimal annotation overhead.

**Strengths:**

1. The paper is well-written and easy to follow;

2. DDCF can leverage unannotated data in SFT;

3. DDCF can balance difficulty and diversity in data selection.

**Weaknesses:**

1. Considering that $h_m$ in the model encoder does not change the dimension of the embeddings, why should the authors use $h_m$ in the formulation of the model encoder, please provide some evidence to demonstrate the necessity of $h_m$.

2. For the experimental setting, [1] has demonstrated that most data selection algorithms can not outperform random selection in the experiments on datasets with more than 1 million data points regarding both effectiveness and efficiency. Data selection algorithms are expected to be applied in real-world industry scenarios, which often need to deal with datasets containing more than 1 million data points. Therefore, empirical results only on OpenR1-Math-220K can not provide convincing support to demonstrate the effectiveness of DDCF for real-world scenarios.

3. The provided empirical results are not that promising, some results can not even outperform the base model, like GSM8k in Table. 2,

4. Considering that the proposed DDCF needs several LLMs to conduct annotation before conducting data selection, and it will obviously add computational cost, the authors should provide computational anaylsis in the paper to demonstrate the efficiency of the proposed method.

[1] Xia, T., Yu, B., Dang, K., Yang, A., Wu, Y., Tian, Y., ... & Lin, J. (2024). Rethinking data selection at scale: Random selection is almost all you need. arXiv preprint arXiv:2410.09335.

**Questions:**

Please see the weaknesses.

---

> ### Author Response · Authors · 2025-11-21
> **Response to Reviewer ksTm (Part 1/3)**
>
> Dear Reviewer ksTM,
>
> We are grateful for your insightful comments and the care you put into reviewing our submission. Your feedback is highly appreciated. Our responses to the raised concerns are as follows.
>
> > **Weakness 1:** Concerns about the unnecessity of $h_M$.
>
> Thank you for pointing this out. Originally, we included $h_M$ mainly to maintain architectural symmetry between the model and question encoders, which keeps the correctness predictor formulation clean and uniform. To assess whether $h_M$ is actually needed, we retrained the correctness predictor with $h_M$ and saw that removing $h_M$ changes the accuracy slightly 80.3% to 80.4% (on 1,947 unseen questions). We will take your suggestion to remove this component later.

---

> ### Author Response · Authors · 2025-11-21
> **Response to Reviewer ksTm (Part 2/3)**
>
> > **Weakness 2:**  Concerns about the effectiveness of Random Selection in >1M-example corpora [1].
>
> We appreciate the comparison with [1]. Our results complement theirs by highlighting two key distinctions:
> - **DDCF consistently outperforms random selection.** Even though random sampling is a strong baseline, DDCF yields clear improvements. For example, with Qwen2.5-Math-7B (Table 2), DDCF surpasses random selection by +10.4 on AIME24, +2.4 on ID average, and +4.9 on OOD average.
> -  **Random selection behaves very differently depending on corpus quality.** The corpora used in [1]—OpenHermes 2.5 and WildChat-1M—are large but relatively redundant. This redundancy explains why their fine-tuning on the full dataset yields only ~1% improvement over training on a 1% random subset: the datasets contain many near-duplicate or easy examples, so a random mini-subset is already highly representative.
>
>     In contrast, OpenR1-Math-220K contains fine-grained and highly varied mathematical problems, from high-school to Olympiad level. For such data, the full corpus offers up to an 8% ID improvement over random selection (Tables 2 and 8).
>
> In summary, we argue that the effectiveness of sophisticated data selection methods over random selection is determined more by corpus quality than corpus size. If the corpus $\mathcal{D}$ lacks meaningful variation in example difficulty—e.g., dominated by overly easy or homogeneous problems—then no selection algorithm can extract a high-value subset, and the downstream performance will inevitably plateau.
>
> To illustrate this, we construct a synthetic “bad” corpus $\mathcal{D}^{\text{syn}}$ containing mostly easy and homogeneous questions. From the remaining pool, we intentionally sample 19,000 "bad" items (maximizing the composite score of Equation 5 in our paper) and 1,000 “cherry” items (minimizing the score of Equation 5), producing a highly skewed dataset where only 5% of random samples would be truly high-quality. The evaluation results are shown below.
>
> ---
> **Rebuttal Table 8:** Performance on In-Distribution and Out-of-Distribution benchmarks.
>
> **Qwen2.5-Math-7B**
> | **Method**  | AIME24 | MATH | OlyBen | GSM8k |  SAT | **ID Avg.** | Minerva | Gaokao | STEM | **OOD Avg.** |
> | ----------- | -----: | ---: | -----: | ----: | ---: | ----------: | ------: | -----: | ---: | -----------: |
> | Random_Low  |   39.6 | 72.9 |   30.6 |  91.4 | 96.8 |        66.3 |    40.1 |   68.6 | 73.8 |         63.2 |
> | Random_High |   38.6 | 76.4 |   34.8 |  91.0 | 98.2 |        67.8 |    41.2 |   64.6 | 75.7 |         60.5 |
> | **DDCF**    |   49.0 | 77.6 |   35.0 |  91.2 | 98.2 |        70.2 |    45.6 |   74.7 | 75.8 |         65.4 |
>
>
> **Qwen3-8B-Base**
> | **Method**  | AIME24 | MATH | OlyBen | GSM8k |  SAT | **ID Avg.** | Minerva | Gaokao | STEM | **OOD Avg.** |
> | ----------- | -----: | ---: | -----: | ----: | ---: | ----------: | ------: | -----: | ---: | -----------: |
> | Random_Low  |   72.1 | 89.0 |   54.8 |  96.4 | 98.6 |        82.2 |    63.0 |   80.0 | 90.2 |         77.7 |
> | Random_High |   80.9 | 90.4 |   53.8 |  94.4 | 99.6 |        83.6 |    62.5 |   83.5 | 90.8 |         79.0 |
> | **DDCF**    |   82.2 | 91.0 |   56.0 |  95.9 |  100 |        85.0 |    66.2 |   84.8 | 90.6 |         80.5 |
>
>
> **Falcon-10B-Base**
> | **Method**  | AIME24 | MATH | OlyBen | GSM8k |  SAT | **ID Avg.** | Minerva | Gaokao | STEM | **OOD Avg.** |
> | ----------- | -----: | ---: | -----: | ----: | ---: | ----------: | ------: | -----: | ---: | -----------: |
> | Random_Low  |   60.2 | 81.4 |   42.1 |  93.8 | 97.3 |        75.0 |    57.2 |   72.1 | 89.6 |         73.0 |
> | Random_High |   65.5 | 82.2 |   47.0 |  93.1 | 98.6 |        77.3 |    58.1 |   79.8 | 89.2 |         75.7 |
> | **DDCF**    |   66.6 | 83.0 |   46.1 |  93.9 | 98.1 |        77.6 |    60.3 |   78.5 | 88.9 |         75.9 |
>
> Across Qwen2.5-Math-7B, Qwen3-8B-Base, and Falcon-10B-Base, **Random_Low** (random selection in bad corpus $\mathcal{D}^{\text{syn}}$) significantly performs worse than **Random_High** (random in full corpus $\mathcal{D}$ OpenR1-Math-220K) on challenging benchmarks such as AIME24, MATH, and OlympiadBench—e.g., gaps of −3.3 (MATH) and −4.2 (OlyBen) for Qwen2.5-Math-7B, −8.2 (AIME24) for Qwen3-8B-Base, and −5.3 (AIME24) for Falcon-10B-Base. Therefore, while LLMs can learn easy tasks without curated data, sophisticated selection methods are essential for challenging problems. Random selection from an easy-heavy corpus is hard to yield expert-level performance (e.g., AIME24).
>
> Finally, this empirical observation aligns with recent theory [2], which proves that under a strong base model and a corpus rich in hard examples, "keep-hard" selection is provably advantageous - showing notable gains over random.
>
> ### **References**
> 1. Xia, et al. "Rethinking Data Selection at Scale: Random Selection Is Almost All You Need." Findings of EMNLP, 2025.
> 2. Dohmatob, el al. *Why Less Is More (Sometimes): A Theory of Data Curation.* arXiv preprint (2025).

---

> > ### Author Response · Authors · 2025-11-21
> > **Response to Reviewer ksTm (Part 3/3)**
> >
> > > **Weakness 3:** Some results do not outperform the base model (e.g., GSM8K in Table 2).
> >
> > Our benchmarks span easy (GSM8K, SAT) to hard (AIME24, MATH, OlympiadBench) tasks to evaluate robustness across difficulty levels. As shown in Section 4.4 and Figure 4, models optimized solely for difficult examples tend to underperform on easier tasks—indicating a forgetting effect. Unlike prior methods such as LIMO or s1.1-1K, which overly emphasize difficulty, DDCF introduces a controllable Difficulty–Diversity trade-off via the $\lambda$ parameter.
> >
> > Although DDCF with Qwen2.5-Math-7B does not surpass the base model on GSM8K in Table 2, it achieves the best overall performance and the top GSM8K score across all baselines, underscoring the efficacy of the proposed trade-off. Furthermore, DDCF offers flexibility: if the user care more about performance on easy benchmarks like GSM8k, they can easily decrease the weight $\lambda$ at the k-greedy step of Equation (5) without additional retraining or redesigning the framework.
> >
> > > **Weakness 4:** Computational analysis on the efficiency of the proposed DDCF.
> >
> > Thank you for the suggestion to detail the computational cost. To isolate the overhead of each data-selection method, we compare runtime only on pre-annotated corpora such as OpenR1-Math-220K (ignoring annotation cost, where DDCF has a clear advantage). Using 8×H100 GPUs, the runtimes are:
> >
> > ---
> > **Rebuttal Table 1**: Runtime comparisions (using 8xH100 GPUs) of data selection methods.
> > | **Method**      | **DDCF** | **Perplexity** | **Cartography** | **S2L** | **DiSF** | **Random** |
> > | --------------- | -------: | -------------: | --------------: | ------: | -------: | ---------: |
> > | Qwen2.5-Math-7B |      1.4 |            1.3 |             1.3 |     2.0 |      0.1 |        0.0 |
> > | Qwen3-8B-Base   |      1.4 |            1.6 |             1.6 |     2.0 |      0.1 |        0.0 |
> > | Falcon-10B-Base |      1.4 |            2.1 |             2.1 |     2.0 |      0.1 |        0.0 |
> > | **Total**       |  **1.4** |        **5.0** |         **5.0** | **2.0** |  **0.1** |    **0.0** |
> >
> >
> > DDCF requires 1.4 GPU hours total, since all participating models share the same seed-set inference; while Correctness Predictor training and the k-greedy step incur negligible cost. In contrast, Perplexity and Cartography must process the full corpus separately for each model to compute token-level probabilities, accumulating 5.0 hours—over 3x the cost of DDCF. Model-agnostic methods such as S2L and DiSF are faster but produce a single fixed subset, offering no way to tailor data to different models.
> >
> > Finally, Appendix C shows that DDCF remains strong even when using far fewer seed questions or models, confirming that its seed-inference cost is bounded, scalable, and efficient across target LLMs.
> >
> > *Note: Cartography was added as a baseline at the suggestion of Reviewer b2jG.*

---

> > > ### Comment · Reviewer_ksTm · 2025-11-27
> > >
> > > Thank you for your detailed responses to my concerns. W1 and W4 are totally addressed. While for W2 and W3, I still have some concerns.
> > >
> > > For W2, I admit that SFT is heavily reliant on data quality rather than data quantity. However, in practical use, large volume high-quality data are difficult to obtain, while mixed-quality data are usually available for training. Thus, the provided response can not get me convinced because I think the proposed method is expected to be applied in real-world scenarios. To be specific, I expect the authors to provide experimental results on OpenHermes 2.5 or WildChat-1M to show DDCF's applicability to real-world scenarios and give comparisons to random selections.
> > >
> > > For W3, the authors explained that DDCF underperforms on GSM8k due to a forgetting effect. However, as shown in Table 2, the DDCF outperforms the baseline on the SAT dataset, which is also an easy text dataset.
> > >
> > > Considering the above two weaknesses that have not been addressed, I can not update my score.

---

> > > > ### Author Response · Authors · 2025-12-02
> > > > **Final Response to Reviewer ksTm**
> > > >
> > > > Thank you Reviewer ksTm for your reply! Let us address your remaining concerns as follows:
> > > >
> > > > > **Weakness 2:** Concerns about the practicality of OpenR1-Math-220K.
> > > >
> > > > We appreciate your emphasis on real-world applicability, particularly regarding two key aspects: **large-scale datasets** and **mixed-quality data**. We also agree with your acknowledgement that, in SFT, **data quality is often more critical than data quantity**.
> > > >
> > > > Indeed, our paper and rebuttal investigate both of these aspects:
> > > > 1. **Large-scale setting** is addressed via Table 2 and the data size ablation in Section 4.3 of our manuscript.
> > > > 2. **Mixed-quality data** is analyzed through our synthetic experiments (see Rebuttal Table 8, our Response Part 2/3), where we control for quality variation directly.
> > > >
> > > > Our discussion is converging on the same overarching conclusion: **The success of data selection relies more on the presence of high-quality "cherry" examples than on corpus size.**
> > > > Crucially, this insight holds regardless of whether the dataset contains 220K or over 1M examples—if the data lacks meaningful variation in difficulty or quality, no selection strategy can extract a truly valuable subset.
> > > >
> > > > While we acknowledge that OpenHermes-2.5 (4,513 downloads/month) and WildChat-1M (6,635 downloads/month) are widely used, OpenR1-Math-220K (12,043 downloads/month) remains actively adopted by the community. This suggests that OpenR1-Math-220K is also common in real-world scenarios.
> > > > Therefore, our work provides practical and transferable insights for practitioners working with large corpus where data quality is available but constrained by resource budgets.
> > > >
> > > > *Note: the number of downloads/month was recorded by Huggingface on 30 Nov 2025.*
> > > >
> > > > > **Weakness 3:** Concerns about the result of GSM8k of Qwen2.5-Math-7B with DDCF.
> > > >
> > > > To clarify this point, we provide the averaged results over 5 runs in **Rebuttal Table 3**. These results show that DDCF does not underperform on GSM8k; rather, it matches the base model almost exactly.
> > > >
> > > > ---
> > > > **Rebuttal Table 2:** Performance ($\texttt{mean}\pm\texttt{std}$) over 5 runs on In-Distribution and Out-of-Distribution benchmarks.
> > > >
> > > > **Qwen2.5-Math-7B**
> > > > | **Method**  | AIME24          | MATH            | OlyBen          | GSM8k           | SAT             | **ID Avg.**              | Minerva         | Gaokao          | STEM            | **OOD Avg.**             |
> > > > | ----------- | --------------- | --------------- | --------------- | --------------- | --------------- | ------------------------ | --------------- | --------------- | --------------- | ------------------------ |
> > > > | Base Model   |   34.6 | 55.4 |   16.4 |  91.6 | 80.0 |        55.6 |    12.9 |   67.1 | 67.7 |         49.2 |
> > > > | **DDCF**    | $51.6_{\pm2.0}$ | $77.6_{\pm0.5}$ | $34.5_{\pm1.0}$ | $91.5_{\pm0.6}$ | $97.4_{\pm0.5}$ | $70.5_{\pm0.4}$ | $44.1_{\pm2.7}$ | $73.9_{\pm2.8}$ | $75.1_{\pm0.4}$ | $64.4_{\pm1.1}$ |
> > > >
> > > > These results highlight two important points:
> > > >
> > > > 1. **GSM8k performance is already saturated for Qwen2.5-Math-7B.** The base model achieves **91.6**, leaving little headroom for further improvement. DDCF preserves this capability (**91.5**) rather than degrading it, while significantly improving performance on more challenging tasks (e.g., +17.0 on AIME24 and +22.2 on MATH). This matches our goal of avoiding erosion of pre-trained knowledge while enhancing reasoning ability.
> > > >
> > > > 2. **SAT, although easier than AIME/MATH, benefits from targeted elicitation.**
> > > > Qwen2.5-Math-7B does not solve SAT optimally from its pre-trained state (**80.0**). DDCF selects a compact but informative subset that raises SAT to **97.4**, showing that the method improves "easy-but-not-saturated" tasks while maintaining saturated ones such as GSM8k.
> > > >
> > > > Overall, the contrast between GSM8k and SAT does not contradict our earlier explanation. Instead, it illustrates the intended behavior of our Difficulty-Diversity trade-off (Section 4.4): **eliciting the model's full problem-solving ability from a small, curated fine-tuning dataset without eroding its pre-trained knowledge.**

---

### Official Review · Reviewer_KfMD · 2025-11-01

**Soundness:** 3
**Presentation:** 3
**Contribution:** 3
**Rating:** 6
**Confidence:** 5

**Summary:**

This paper proposed Difficulty–Diversity Collaborative Filtering (DDCF) as an algorithm for model-specialized data filtering, which can be applied for LLM finetuning and active learning for CoT trace annotation. From the collaborative filtering, the learned parametrized model embedding and problem encoder can predict the problem's correctness; then the final data selection scores combine the difficulty and diversity of the selected subset, using the k-greedy strategy.
Empirically, the proposed method achieves a SOTA performance, while with 100x less annotation costs.

**Strengths:**

1. The proposed DDCF algorithm is neat and effective, which provides an efficient and reliable method to quantify the model-specialized difficulty and diversity.

2. The DDCF method demonstrates a significant improvements on the data efficiency on both instruction tuning on annotated and unannotated data.

3. The paper is well-written and easy to follow, with clear illustration and extensive experimental results.

**Weaknesses:**

1. The proposed collaborative filtering algorithm might have a cold start problem with insuffient data samples or biased data distributions. The performance can also greatly depend on the seed coreset.

2. Lack of ablations on the number of models and questions used to train the CF model.

**Questions:**

1. Have you encountered any cold start issues when training the collaborative filtering framework? How do you collect the seed dataset used to train the CF system?

2. According to Figure 5, the data distribution selected by different models seems quite close. How much could the collaborative filtering from multiple models improves the performance comparing to train the question encoder with a classifier head upon a single model's correctness?

3. Are there any performance differences when using models from different families/scales? Would the obtained encoder also be able to transferrable between different model scales?

---

> ### Author Response · Authors · 2025-11-21
> **Response to Reviewer KfMD (Part 1/2)**
>
> Dear Reviewer KfMD,
>
> Thank you for your detailed and helpful reviews. We value the effort you have invested in providing such meaningful feedback. We respond to your concerns point by point below.
>
> > **Weakness 1 + Question 1.1:** Cold-start concerns about training the Correctness Predictor with insufficient seed data.
>
> Our results in Table 6 and Table 7 (Appendix C) confirm that the CF model is reliable even with minimal data (for example, 1,000-2,000 data samples) and scales effectively: most gains emerge early, with incremental benefits thereafter. Moreover, cold-start issues were not observed during the training of the Correctness Predictor. This can be attributed to two main factors. First, the framework leverages a pre-trained Sentence-Transformer to encode questions into semantically rich embeddings. These representations provide a warm start, preventing the model from starting with random or uninformative features. Second, the CF model itself is simple and lightweight—comprising only a few MLP layers—which reduces the risk of overfitting or instability.
>
> > **Question 1.2:**  How to collect the seed dataset?
>
> In DDCF, the seed dataset $\mathcal{Q}$ only needs to share the same domain distribution as the large corpus $\mathcal{D}$. In practice, this can be obtained simply by sampling questions directly from $\mathcal{D}$. Importantly, DDCF requires only question–answer pairs to build the binary correctness matrix (Figure 1), the seed dataset is easy to prepare. To keep experiments clear and reproducible, we instead rely on publicly available but domain-aligned datasets: 19,470 questions from GSM8K and MATH for the main experiments (Section 4), and 1,531 validation questions from MMLU for the unannotated-corpus setting (Section 5).
>
> > **Weakness 2 + Question 2.2:** Ablations on the number of models and questions used to train the Correctness Predictor.
>
> ---
> **Rebuttal Table 1:** Effect of Number of Participating Models on ID Performance
>
> | **# Models** | 1    | 2    | 4    | 8    | 16   | 23   |
> | ------------ | ---- | ---- | ---- | ---- | ---- | ---- |
> | **Accuracy** | 69.5 | 70.1 | 69.9 | 69.8 | 69.9 | 70.2 |
> ---
> **Rebuttal Table 2:** Effect of Number of Seeding Questions on ID Performance
>
> | **# Questions** | 1K   | 2K   | 4K   | 8K   | 16K  | 17.5K |
> | --------------- | ---- | ---- | ---- | ---- | ---- | ----- |
> | **Accuracy**    | 69.3 | 68.5 | 69.5 | 70.4 | 70.0 | 70.2  |
> ---
> **Rebuttal Table 3:** Effect of Number of Participating Models on OOD Performance
>
> | **# Models** | 1    | 2    | 4    | 8    | 16   | 23   |
> | ------------ | ---- | ---- | ---- | ---- | ---- | ---- |
> | **Accuracy** | 61.7 | 60.6 | 63.5 | 62.2 | 63.8 | 65.4 |
> ---
> **Rebuttal Table 4:** Effect of Number of Seeding Questions on OOD Performance
>
> | **# Questions** | 1K   | 2K   | 4K   | 8K   | 16K  | 17.5K |
> | --------------- | ---- | ---- | ---- | ---- | ---- | ----- |
> | **Accuracy**    | 63.0 | 62.4 | 61.7 | 63.2 | 64.0 | 65.4  |
>
> Thank you for the suggestion on the ablation studies. We further vary the seed matrix along two axes—(1) fixing 17.5K questions while changing the number of seed models, and (2) fixing 23 models while changing the number of seeding questions—and fine-tune Qwen2.5-Math-7B on the resulting DDCF-selected subsets (Rebuttal Tables 1–4).
>
> - **ID performance is strikingly stable.** Accuracy remains tightly concentrated (68.5–70.4) across all settings, with a mean of 69.8 and a standard deviation of 0.25. This shows that DDCF reliably identifies strong in-domain training samples even with substantially fewer seed models or questions.
>
> -  **OOD performance benefits from scale.** Increasing the number of participating models improves OOD accuracy from 61.7 (1 model) to 65.4 (23 models). Similarly, expanding seeding questions improves OOD accuracy from 63.0 (1K) to 65.4 (17.5K). This shows that larger seed matrices provide richer difficulty signals and improve generalization..
>
> -  **Summary.** ID performance is robust even under very small seed budgets, while OOD performance improves steadily with more seed models and questions. This efficiency–scaling pattern makes DDCF cost-effective for in-domain fine-tuning and scalable when stronger OOD robustness is desired.
>
> We will add this ablation study into the next version of the paper to comprehend the proposed DDCF.

---

> ### Author Response · Authors · 2025-11-21
> **Response to Reviewer KfMD (Part 2/2)**
>
> > **Question 2.1 + Question 3:** Topic distribution of DDCF and ablations on data-transferability between LLMs.
>
> In lines 430-431, we shows that the average intra-family Jaccard index of DDCF-selected subsets is 0.224 versus 0.169 inter-family. Despite close LLMs tend to have similar topic distributions (Figure 5), their actual selected questions overlap by only 22%. This indicates that DDCF does not produce homogeneous subsets but instead tailors its selections to each model’s specific strengths and weaknesses.
>
> Our data-transferability study (Appendix E) further shows that these model-specific subsets can still transfer effectively across LLMs. Transfer is strongest between models with closer architectural or training kinship, suggesting that the correctness-predictor embeddings capture generalizable patterns while remaining sensitive to model-family characteristics.

---

### Author Response · Authors · 2025-12-02
**Summary of the Discussion Period**

Dear Reviewers, Area Chairs, and SACs,

We sincerely thank the reviewers for their thoughtful and constructive feedback. We are encouraged that all reviewers found the paper well-presented and recognized DDCF as a simple yet broadly applicable (KfMD, ksTm, b2jG), computationally efficient (ksTm, b2jG), and empirically well-supported framework (KfMD, b2jG). Reviewers also expressed enthusiasm for the difficulty–diversity trade-off principle (KfMD, ksTm) and appreciated the additional insights enabled by DDCF, such as the performance–cost trade-off and the proportion of "cherry" examples (b2jG).

During the active rebuttal phase, most reviewers responded positively, and nearly all concerns were addressed. Only a few ongoing points remain:
- Reviewer ksTm's comments prompted two aspects of data selection, solved the data-size ablation (Section 4.3) and a new synthetic experiment, and also led to clarifying the forgetting effect underlying the GSM8K dip in Difficulty–Diversity trade-offs.
- Reviewer KfMD's concerns regarding cold-start and data-transferability were resolved through further explanation and the ablation in Appendix E.

We regret that an OpenReview identity issue disrupted an otherwise productive rebuttal. For instance, Reviewer b2jG had already been fully convinced and raised their score from 6 to 8 before the incident.

We have revised the manuscript according to the suggestions (changes highlighted in blue). Updates include: expanded related work in Section 2 (per b2jG); refinements to the Correctness Predictor's architecture in Section 3.1 (per ksTm); improved writing and incorporation of Dataset Cartography as a new baseline in Tables 2, 14 (per b2jG); and a detailed discussion of corpus quality in Section 4.6 (per ksTm and b2jG). Further ablations on performance–cost trade-off appear in Appendix C.2 (per KfMD, b2jG), and computational analysis in Appendix B (per ksTm, bj2G).

We hope positive discussions and score progression before the incident can be taken into account in the final decision. Thank you for your time and understanding.

Regards,

Authors of Paper 15954

---

### Meta-Review · Area_Chair_J95y · 2026-01-14

**Summary:**

The paper proposes Difficulty–Diversity Collaborative Filtering (DDCF), a model-specialized data selection method: learn a lightweight correctness predictor from a model–question correctness matrix, then build a training set by balancing predicted difficulty and semantic diversity. Experiments on OpenR1-Math-220K across several target LLMs show consistent gains over random and other selection baselines.

The main concerns raised in reviews were:
- potential cold-start / seed-bias and transferability across model families (KfMD)
- need for ablations on number of seed models/questions and computational cost (KfMD, ksTm, b2jG),
- questions about real-world scalability vs the “random is enough at 1M+” narrative and mixed-quality corpora (ksTm)
- missing related work baselines (Datamodels / Dataset Cartography) plus multi-run robustness and clearer claims around where DDCF is not best (b2jG).


The key claims (effective, simple difficulty–diversity selection; robustness to seed size; competitive runtime/cost) are supported in the paper and strengthened by the rebuttal. The remaining concerns are minor and cannot be seen as a flaw in the method or the experiments.

**Reviewer Concerns:**

# KfMD

Mostly addressed:
- Cold-start/seed-bias concerns are mitigated by reported robustness with small seed budgets (Appendix C) and the warm-start question embeddings
- Ablations directly vary #seed models/questions and show improved OOD as seed scale grows.
-Transferability/cross-family concerns are partly addressed by overlap analysis (Sec. 4.5) and the explicit transfer experiments in Appendix E.


# ksTm
Mostly addressed, with one outstanding external-validity request.
- architectural component concern resolved via an ablation and planned simplification
- efficiency concern alleviated via a runtime comparison showing competitive total cost (Table 7).
- Remaining concern: reviewer asks for experiments on OpenHermes/WildChat-1M to demonstrate applicability to large mixed-quality corpora; these are not in the paper, so this point remains open.
- The GSM8K/SAT discussion is clarified via saturation/forgetting and the tunable trade-off (Sec. 4.4).

#b2jG
Addressed: Authors add/position related work (Datamodels, Dataset Cartography), include Cartography as a baseline, provide cost–performance discussion/runtime, report multi-run results, and soften “always best” phrasing.

**Reviewer Scores:**

KfMD: 6 -> 6 or 8  : main weaknesses directly answered by ablations + transfer analysis).
ksTm: 4 -> 4 : explicitly states remaining concerns)
b2jG: 6 ->8 : explicit score increase after rebuttal).

---

### Decision · Program_Chairs · 2026-01-26

Accept (Poster)